# Unitary convolutions for learning on graphs and groups

Bobak T. Kiani[*]    Lukas Fesser[†]    Melanie Weber[‡]

Data with geometric structure is ubiquitous in machine learning often arising from fundamental symmetries in a domain, such as permutation-invariance in graphs and translation-invariance in images. Group-convolutional architectures, which encode symmetries as inductive bias, have shown great success in applications, but can suffer from instabilities as their depth increases and often struggle to learn long range dependencies in data. For instance, graph neural networks experience instability due to the convergence of node representations (over-smoothing), which can occur after only a few iterations of message-passing, reducing their effectiveness in downstream tasks. Here, we propose and study *unitary group convolutions*, which allow for deeper networks that are more stable during training. The main focus of the paper are graph neural networks, where we show that unitary graph convolutions provably avoid over-smoothing. Our experimental results confirm that unitary graph convolutional networks achieve competitive performance on benchmark datasets compared to state-of-the-art graph neural networks. We complement our analysis of the graph domain with the study of general unitary convolutions and analyze their role in enhancing stability in general group convolutional architectures.[4]

## 1  Introduction

In recent years, the design of specialized machine learning architectures for structured data has received a surge of interest. Of particular interest are architectures for data domains with inherent symmetries, such as permutation-invariance in graphs and sets, translation-invariance in images, and other symmetries that arise from fundamental laws of physics in scientific data.

Group-convolutional architectures allow for explicitly encoding symmetries as inductive biases, which has led to performance gains in scientific applications [ESM23, WWY20]; theoretical studies have analyzed the impact of geometric inductive biases on the complexity of the architecture [MMM21, BVB21, KLL+24]. Graph Neural Networks are among the most popular architectures for graph machine learning and have found impactful applications in a wide range of disciplines, including in chemistry [GRK+21], drug discovery [ZAL18], particle physics [SBV20], and recommender systems [WSZ+22]. However, despite these successes, several limitations remain. A notable difficulty is the design of stable, deep architectures. Many of the aforementioned applications require accurate learning of long-range dependencies in the data, which necessitates deeper networks. However, it has been widely observed that group-convolutional networks suffer from instabilities as their depths increases. On graph domains, these instabilities have been studied extensively in recent years, notably in the form of *over-smoothing* effects [RBM23], which characterizes the fast convergence of the representations of nearby nodes with depth. This effect can often be observed after only a few iterations of message-passing (i.e., small number of layers) and can significantly decrease the utility of the learned representations in downstream tasks. While interventions for mitigating over-smoothing have been proposed, including targeted perturbations of the input graph's connectivity (rewiring) and skip connections, a more principled architectural approach with theoretical guarantees is still

---

[*]John A. Paulson School of Engineering and Applied Sciences, Harvard; e-mail: `bkiani@g.harvard.edu`
[†]John A. Paulson School of Engineering and Applied Sciences, Harvard; e-mail: `lukas_fesser@fas.harvard.edu`
[‡]John A. Paulson School of Engineering and Applied Sciences, Harvard; e-mail: `mweber@g.harvard.edu`

[4]Code available at `https://github.com/Weber-GeoML/Unitary_Convolutions`

38th Conference on Neural Information Processing Systems (NeurIPS 2024).

lacking. Similar effects, such as exploding or vanishing gradients, have also been studied in more general group-convolutional architectures, specifically in CNNs [TK21, SF21, LJW+19], for which architectural interventions (e.g., skip connections) have been proposed.

In this work, we take a different route. Inspired by a long line of work studying unitary recurrent neural networks [ASB16, HSL16, LCMR19, KBLL22], we propose to replace the standard group convolution operator with a *unitary* group convolution. By construction, the unitarity ensures that the linear transformations are norm-preserving and invertible, which can significantly enhance the stability of the network and avoid convergence of representations to a fixed point as its depth increases. We introduce two unitary graph convolution operators, which vary in the way the message passing and feature transformation are parameterized. We then generalize this approach to cover more general group-convolutional architectures.

Our theoretical analysis of the proposed unitary graph convolutions shows that they enhance stability and prevent over-smoothing effects that decrease the performance of their vanilla counterparts. We further describe how generalized unitary convolutions avoid vanishing and exploding gradients, enhancing the stability of group-convolutional architectures without additional interventions, such as residual connections or batch normalization.

## 1.1 Related work

We now provide a brief background into work that motivated and inspired this current study, deferring a more complete discussion to App. A. Unitary matrices have a long history of application in neural networks, specifically related to improving stability for deep networks [SMG13, PMB13] enhancing the learning of long-range dependencies in data [BSF94, Hoc91, SMG13]. [ASB16, JSD+17, HSL16] implemented unitary matrices in recurrent neural networks to address issues with the challenge of vanishing and exploding gradients inherent to learning long sequences of data in RNNs [BSF94, LJH15]. These original algorithms were later improved to be more expressive while still being efficient to implement in practice [HWY18, LCMR19, KBLL22]. For graph convolution, [HSTW20] discuss applications of the exponential map to linear convolutions; here, we use this same exponential map to explicitly apply unitary operators parameterized in the Lie algebra. For image data, various works [SGL18, LHA+19, TK21, SF21, KBLL22] design and analyze variants of orthogonal or unitary convolution used in CNN layers. These can be viewed as a particular instance of the group convolution we study here over the cyclic group. More recently, proposals for unitary or orthogonal message passing have shown improvements in stability and performance compared to conventional message passing approaches [GZH+22, AEL+24, QBY24]. However, in contrast to our work, these methods do not always implement a unitary transformation across the whole input (e.g. only applying it in the feature transformation) and in the case of [QBY24] can be computationally expensive to implement for large graphs (see App. A for more detail).

## 2 Background and Notation

We denote scalars, vectors, and matrices as $c, \boldsymbol{w}$, and $\boldsymbol{M}$ respectively. Given a matrix $\boldsymbol{M}$, its conjugate transpose and transpose are denoted $\boldsymbol{M}^\dagger$ and $\boldsymbol{M}^\top$. Given two matrices, $\boldsymbol{A}$ and $\boldsymbol{B}$, we denote their tensor product (or Kronecker product) as $\boldsymbol{A} \otimes \boldsymbol{B}$. Given a vector $\boldsymbol{w}$, its standard Euclidean norm is denoted $\|\boldsymbol{w}\|$. For a matrix $\boldsymbol{M}$, we denote its operator norm as $\|\boldsymbol{M}\|$ and Frobenius norm as $\|\boldsymbol{M}\|_F$.

**Group Theory Basics** Symmetries ("invariances") describe transformations, which leave properties of data unchanged ("invariant"), and as such characterize the inherent geometric structure of the data domain. Algebraically, symmetries can be characterized as groups. We say that a group is a matrix *Lie group*, if it is a differentiable manifold and a subgroup of the set of invertible $n \times n$ matrices (see App. C). Lie groups are associated with a Lie algebra, a vector space, which is formed by its tangent space at the identity. A comprehensive introduction into Lie groups and Lie algebras can be found in [FH13, Hal15]. Throughout this work we will encounter the $n$-dimensional orthogonal $O(n)$ and unitary $U(n)$ Lie groups, which are defined as

$$O(n) = \left\{ \boldsymbol{U} \in \mathbb{R}^{n \times n} : \boldsymbol{U}\boldsymbol{U}^\top = \boldsymbol{I} \right\}, \qquad U(n) = \left\{ \boldsymbol{U} \in \mathbb{C}^{n \times n} : \boldsymbol{U}\boldsymbol{U}^\dagger = \boldsymbol{I} \right\}. \tag{1}$$

Lie algebras of $O(n)$ and $U(n)$ are the set of skew symmetric $\mathfrak{o}(n)$ and skew Hermitian $\mathfrak{u}(n)$ matrices respectively, i.e.,

$$\mathfrak{o}(n) = \left\{ M \in \mathbb{R}^{n \times n} : M + M^\top = 0 \right\}, \qquad \mathfrak{u}(n) = \left\{ M \in \mathbb{C}^{n \times n} : M + M^\dagger = 0 \right\}. \tag{2}$$

Given a matrix $M \in \mathfrak{o}(n)$ (or $\mathfrak{u}(n)$), the matrix exponential maps the matrix to an element of the Lie group $\exp(M) \in O(n)$ (or $U(n)$). More details on unitary parametrizations can be found in App. C.2.

**Graph Neural Networks** We denote graphs by $\mathcal{G} = (V, E)$ where $V$ and $E$ denote the set of nodes and edges respectively. For a graph on $n$ nodes, unless otherwise specified, we let $V = \{1, \ldots, n\}$ index the nodes and denote the adjacency matrix by $A \in \mathbb{R}^{n \times n}$ and node features $x \in \mathbb{R}^{n \times d}$. We also often use $D \in \mathbb{R}^{n \times n}$ to denote the diagonal degree matrix where diagonal entry $i$ records the degree of node $i$. The normalized adjacency matrix is defined as $\widetilde{A} = D^{-1/2} A D^{-1/2}$. Given a node feature matrix $X \in \mathbb{R}^{n \times d_{\text{in}}}$ where row $i$ denotes the $d_{\text{in}}$-dimensional feature vector of node $i$, graph convolution operators take the general form [KW16, ZK20]

$$f_{\text{conv}}(X; A) = X W_0 + A X W_1 + \cdots + A^k X W_k, \tag{3}$$

where $W_0, \ldots W_k \in \mathbb{R}^{d_{\text{in}} \times d_{\text{out}}}$ are trainable parameters. For ease of presentation, we will often omit the adjacency matrix as an explicit input to the operation. Often, only a single "message passing" step is included and the operation takes the simple form $f_{\text{conv}}(X) = A X W$. Eq. (3) is equivariant under any permutation matrix $P_\pi \in \mathbb{R}^{n \times n}$[5] since

$$f_{\text{conv}}(P_\pi X; P_\pi A P_\pi^{-1}) = P_\pi \cdot f_{\text{conv}}(X; A). \tag{4}$$

We aim to parameterize a particular subset of these operations which preserve unitarity properties.

**Group-Convolutional Neural Networks** In general, a linear convolution operation $\text{conv}_G : \mathbb{C}^{n \times c} \to \mathbb{C}^{n \times c}$ takes the form of a weighted sum over linear transformations that are equivariant to a given group $G$. For simplicity, we assume here that we are working with finite groups though this can be generalized to other settings [KT18, CGKW18, CW16]. Given an input $X \in \mathbb{C}^{n \times c}$ consisting of $c$ channels in a vector space of dimension $n$, we study convolutions of the form

$$\text{conv}_G(X) = \sum_{i=1}^{m} T_i X W_i, \tag{5}$$

where $T_1, \ldots, T_m \in \mathbb{C}^{n \times n}$ are linear operators equivariant to the group $G$ and $W_1, \ldots, W_m \in \mathbb{C}^{c \times c}$ are parameterized weight matrices. The graph setting is recovered by setting $T_k = A^k$. Similarly, for cyclic convolution as in conventional CNNs, one sets $T_k$ to be the circulant matrices sending basis vector $T_k e_i = e_{i+k}$ where indexing is taken mod $n$.

# 3 Unitary Group Convolutions

We first describe unitary convolution for data on graphs (equivariant to permutations) which is the main focus of our study and then detail general procedures for performing unitary convolutions equivariant to general finite groups. Implementing these operations often requires special considerations to handle nonlinearities, complex numbers, initialization, etc. which we discuss in App. E.

## 3.1 Unitary graph convolution

We introduce two variants of unitary graph convolution, which we denote as *Separable unitary convolution* (short *UniConv*) and *Lie orthogonal/unitary convolution* (short *Lie UniConv*). UniConv is a simple adjustment to standard message passing and treats linear transformations over nodes and features separately. Lie UniConv, in contrast, parameterizes operations in the Lie algebra of the orthogonal/unitary groups. This operation is fully unitary, but does not have the tensor product nature of the separable UniConv.

By introducing complex numbers, we can enforce unitarity separately in the message passing and feature transformation.

---

[5]In matrix form, entry $[P_\pi]_{ij}$ is one if $\pi(j) = i$ and zero otherwise.

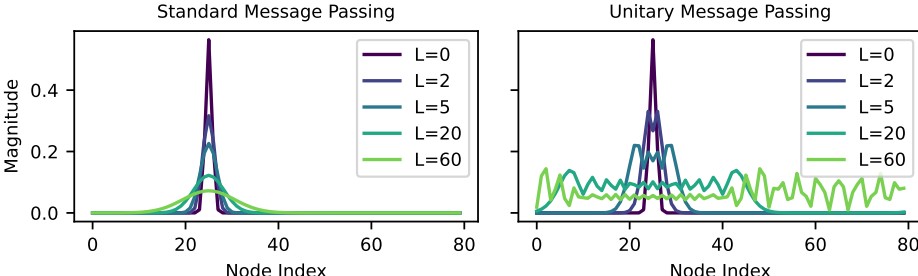

Figure 1: Comparison of standard linear message passing with iterates $x_{L+1} = c(x_L + A x_L)$ versus unitary message passing with iterates $x_{L+1} = \exp(iA)x_L$ for a graph of 80 nodes connected as a ring. The unitary message passing has a wave-like nature which ensures messages "propagate" through the graph. In contrast, the standard message passing has a unique fixed point corresponding to the all ones vector which inherently causes oversmoothing in the features. Here, $c$ is chosen to ensure the operator norm of the matrix $I + A$ is bounded by one.

**Definition 1** (Separable unitary graph convolution (UniConv))**.** Given an undirected graph $\mathcal{G}$ over $n$ nodes with adjacency matrix $A \in \mathbb{R}^{n \times n}$, separable unitary graph convolution (UniConv) $f_{\text{Uconv}}$ : $\mathbb{C}^{n \times d} \to \mathbb{C}^{n \times d}$ takes the form

$$f_{\text{Uconv}}(X) = \exp(iAt)XU, \quad UU^{\dagger} = I, \tag{6}$$

where $U \in U(d)$ is a unitary operator and $t \in \mathbb{R}$ controls the magnitude of the convolution.

One feature of the complexification of the adjacency matrix is that messages propogate as "waves" as observed for example in Fig. 1. Since $A$ is a symmetric matrix, $\exp(iAt)$ is unitary for all values of $t \in \mathbb{R}$ and corresponds to vanilla message passing up to first order: $\exp(iAt) \approx I + iAt + O(t^2)$.

**Remark 2.** We observe that performance on real-world tasks is usually improved when enforcing unitarity in the node message passing where oversmoothing occurs, but not necessarily when enforcing unitarity in the feature transformation $U$. Thus, one can choose to leave $U$ in Eq. (6) as a fully parameterized (unconstrained) matrix as we often do in our experiments.

More generally, one can parameterize the operation in the Lie algebra by first forming a skew Hermitian convolution operation $g_{\text{conv}}$ : $\mathbb{C}^{n \times d} \to \mathbb{C}^{n \times d}$ and then applying the exponential map. This approach has the benefit that it can be fully implemented using real numbers to obtain an orthogonal operator by enforcing constraints in the real part of the weight matrix $W$ only.

**Definition 3** (Lie orthogonal/unitary graph convolution (Lie UniConv))**.** Given an undirected graph $\mathcal{G}$ over $n$ nodes with adjacency matrix $A \in \mathbb{R}^{n \times n}$, Lie unitary/orthogonal graph convolution (Ortho-Conv) $f_{\text{Uconv}}$ : $\mathbb{C}^{n \times d} \to \mathbb{C}^{n \times d}$ takes the form

$$f_{\text{Uconv}}(X) = \exp(g_{\text{conv}})(X) = \sum_{k=0}^{\infty} \frac{g_{\text{conv}}^{(k)}(X)}{k!} = X + g_{\text{conv}}(X) + \frac{1}{2} g_{\text{conv}}(g_{\text{conv}}(X)) + \cdots, \tag{7}$$

$$g_{\text{conv}}(X) = AXW, \quad W + W^{\dagger} = 0.$$

The operation of $g_{\text{conv}}$ can be represented as a vector-matrix operation $\text{vec}(g_{\text{conv}}(X)) = A \otimes W^{\top} \text{vec}(X)$ where $A \otimes W^{\top}$ belongs in the Lie algebra. If $W$ is real-valued, the above returns an orthogonal map since the exponential of a real-valued matrix is real-valued.

**Implementing the exponential map** The exponential map in Definitions 1 and 3 can be performed using accurate approximations with typically constant factor overhead in runtime. We use the simple $K$-th order Taylor approximation

$$\exp(M) = \sum_{k=0}^{K} \frac{M^k}{k!} + O\left(\frac{\|M\|^{K+1}}{(K+1)!}\right). \tag{8}$$

For experiments, we find that setting $K = 10$ suffices in all cases as the error exponentially decreases with $K$. Various other accurate and efficient approximations exist as detailed in App. C.2. We also

refer the reader to App. E for other implementation details associated to handling complex numbers, initialization, etc.

## 3.2 Generalized unitary convolutions

In the more general setting, we are concerned with parameterizing unitary operations of the form

$$\text{conv}_G(X) = \sum_{i=1}^{m} T_i X W_i, \tag{9}$$

where $T_1, \ldots, T_m \in \mathbb{C}^{n \times n}$ are linear operators equivariant to the group $G$ and $W_1, \ldots, W_m \in \mathbb{C}^{c \times c}$ are parameterized weight matrices (e.g., set $T_k = A^{k-1}$ to recover graph convolution). One can enforce and parameterize unitary convolutions in Eq. (9) in the Lie algebra basis or in the Fourier domain as detailed in App. D.

---

**Algorithm 1** Unitary map from Lie algebra

---

**Input:** equivariant linear operator $L \in \mathbb{C}^n \to \mathbb{C}^n$
**Input:** vector $\mathbf{x} \in \mathbb{C}^n$
 1: $\widetilde{L} = \frac{1}{2}(L - L^\dagger)$ (skew symmetrize operator)
 2: **return** $\exp(\widetilde{L})(\mathbf{x})$ (or approximation thereof)

---

In the Lie algebraic setting (Algorithm 1), one explicitly parameterizes operators in the Lie algebra or orthogonally projects arbitrary linear operators onto this basis by the mapping $X \mapsto (X - X^\dagger)/2$. This parameterization is particularly simple to implement (it is a linear basis) and unitary operators are subsequently implemented by applying the exponential map. This setting covers previous implementations of unitary RNNs and CNNs [LCMR19, SF21] and is detailed in Sec. 3.1 for GNNs.

**Example 1** (Convolution on regular representation (Lie algebra)). Given a group $G$ and vector space $\mathcal{V}$ of dimension $|G|$ with basis $\{e_g : g \in G\}$, then the left action $T_g$ (right action $R_g$) of any $g \in G$ is a permutation $T_g e_h = e_{g^{-1}h}$ ($R_g e_h = e_{hg}$). Let $x \in \mathbb{C}^{|G|}$ be the vectorized form of an input function $x : G \to \mathbb{C}$ and $m : G \to \mathbb{C}$ the filter for convolution[6]

$$(m \star x)(u) = \sum_{v \in G} m(u^{-1}v)x(v) \iff \text{conv}_G(x) = \left[\sum_{g \in G} m(g) R_g\right] x. \tag{10}$$

Parameterizing operations on the Lie algebra simply requires that $m(g) = m(g^{-1})^*$ since $R_g^{-1} = R_g^\dagger$.

An example implementation of the above for a toy learning task on the dihedral group is in App. F.3. One can also generally implement convolutions in the (block diagonal) Fourier basis of the graph or group (see App. D and Algorithm 3). Here, one employs a Fourier operator which block diagonalizes the input into its irreducible representations or some spectral representation. Fourier representations often have the advantage of being faster to implement due to efficient Fourier transforms. Since we do not use this in our experiments, we defer the details to App. D.

## 4   Properties and theoretical guarantees

Unitary operators are now well studied in the context of neural networks and have various properties that are useful in naturally enhancing stability and performance of learning architectures [ASB16, SMG13, LCMR19, JSD+17, KBLL22]. These properties and their theoretical guarantees are outlined here. We defer all proofs to App. B, many of which follow immediately from the definition of unitarity.

Throughout, we will assume that convolution operators act on a vector space $\mathcal{V}$ and are built from a basis of linear operators that is equivariant to input and output representation $\rho(g)$ of a group $G$. We set input/output representations to be equal so that the exponential map of an equivariant operator is itself equivariant.

---

[6]Typically called cross-correlation in mathematics literature.

**Fact 1** (Basic properties). *Any unitary convolution $f_{\text{Uconv}} : \mathcal{V} \to \mathcal{V}$ built from Algorithms 1 and 3 meets the basic properties:*

$$
\begin{aligned}
\textit{(invertibility):} \quad & \exists\, f_{\text{Uconv}}^{-1} : \mathcal{V} \to \mathcal{V} \text{ such that } \forall \boldsymbol{x} \in \mathcal{V} : f_{\text{Uconv}}^{-1}(f_{\text{Uconv}}(\boldsymbol{x})) = \boldsymbol{x}, \\
\textit{(isometry):} \quad & \forall \boldsymbol{x} \in \mathcal{V} : \|f_{\text{Uconv}}(\boldsymbol{x})\| = \|\boldsymbol{x}\|, \\
\textit{(equivariance):} \quad & \rho(g) \circ f_{\text{Uconv}} = f_{\text{Uconv}} \circ \rho(g).
\end{aligned}
\tag{11}
$$

A simple corollary of the above isometry property leveraged in prior work on unitary CNNs [TK21, SGL18] is that unitary matrices naturally provide robustness guarantees and a provable means to bound the effects of adversarial perturbations (see Corollary 9 in App. B). For graphs, we note that the properties above are generally impossible to obtain with graph convolution operations that perform a single message passing step as we show below.

**Proposition 4.** *Let $f_{\text{conv}} : \mathbb{R}^{n \times d} \to \mathbb{R}^{n \times d}$ be a graph convolution layer of the form*

$$
f_{\text{conv}}(\boldsymbol{X}, \boldsymbol{A}) = \boldsymbol{X}\boldsymbol{W}_0 + \boldsymbol{A}\boldsymbol{X}\boldsymbol{W}_1,
\tag{12}
$$

*where $\boldsymbol{W}_0, \boldsymbol{W}_1 \in \mathbb{R}^{d \times d}$ are parameterized matrices. The linear map $f(\cdot, \boldsymbol{A}) : \mathbb{R}^{n \times d} \to \mathbb{R}^{n \times d}$ is orthogonal for all adjacency matrices $\boldsymbol{A}$ of undirected graphs only if $\boldsymbol{W}_1 = \boldsymbol{0}$ and $\boldsymbol{W}_0 \in O(d)$ is orthogonal. Furthermore, denoting $\boldsymbol{J}_{\boldsymbol{A}} \in \mathbb{R}^{nd \times nd}$ as the Jacobian matrix of the map $f_{\text{conv}}(\cdot, \boldsymbol{A})$, for any choice of $\boldsymbol{W}_0, \boldsymbol{W}_1$, there always exists a normalized adjacency matrix $\hat{\boldsymbol{A}}$ such that*

$$
\left\| \boldsymbol{J}_{\hat{\boldsymbol{A}}}^{\top} \boldsymbol{J}_{\hat{\boldsymbol{A}}} - \boldsymbol{I} \right\| \geq \frac{\|\boldsymbol{W}_1\|_F^2}{2d},
\tag{13}
$$

*where $\|\boldsymbol{M}\|$ is the operator norm of matrix $\boldsymbol{M}$.*

The above shows that one must apply higher powers of $\boldsymbol{A}$ as in the exponential map to achieve a linear operator that is close to orthogonal.

**Oversmoothing**   During the training of GNNs we often observe that the features of neighboring nodes become more similar as the depth of the networks (i.e., the number of message-passing iterations) increases. This "oversmoothing" phenomenon has a strong connection to the spectral properties of graphs where convergence of a function on the graph is measured through the Dirichlet form[7] or its normalized variant also termed the Rayleigh quotient.

**Definition 5** (Rayleigh quotient [Chu97]). Given an undirected graph $\mathcal{G} = (V, E)$ on $|V| = n$ nodes with adjacency matrix $\boldsymbol{A} \in \{0, 1\}^{n \times n}$, let $\boldsymbol{D} \in \mathbb{R}^{n \times n}$ be a diagonal matrix where the $i$-th entry $\boldsymbol{D}_{ii} = d_i$ and $d_i$ is the degree of node $i$. Let $f : V \to \mathbb{C}^d$ be a function from nodes to features. Then the Rayleigh quotient $R_{\mathcal{G}}(f)$ is equal to

$$
R_{\mathcal{G}}(f) = \frac{1}{2} \frac{\sum_{(u,v) \in E} \left\| \frac{f(u)}{\sqrt{d_u}} - \frac{f(v)}{\sqrt{d_v}} \right\|^2}{\sum_{w \in V} \|f(w)\|^2} = \frac{\operatorname{Tr}\left( \boldsymbol{X}^{\dagger}(\boldsymbol{I} - \widetilde{\boldsymbol{A}})\boldsymbol{X} \right)}{\|\boldsymbol{X}\|_F^2},
\tag{14}
$$

where $\widetilde{\boldsymbol{A}} = \boldsymbol{D}^{-1/2}\boldsymbol{A}\boldsymbol{D}^{-1/2}$ is the normalized adjacency matrix and $\boldsymbol{X} \in \mathbb{C}^{n \times d}$ is a matrix with the $i$-th row set to feature vector $f(i)$. We will at times abuse notation and let $\boldsymbol{X}$ be an input to $R_{\mathcal{G}}(\boldsymbol{X})$.

Given an undirected graph $\mathcal{G}$ on $n$ nodes with normalized adjacency matrix $\widetilde{\boldsymbol{A}} = \boldsymbol{D}^{-1/2}\boldsymbol{A}\boldsymbol{D}^{-1/2}$, we compare the Rayleigh quotient of normalized vanilla and unitary convolution. First, it is straightforward to show that the Rayleigh quotient is invariant to unitary transformations giving a proof that unitary graph convolution avoids oversmoothing.

**Proposition 6** (Invariance of Rayleigh quotient). *Given an undirected graph $\mathcal{G}$ on $n$ nodes with normalized adjacency matrix $\widetilde{\boldsymbol{A}} = \boldsymbol{D}^{-1/2}\boldsymbol{A}\boldsymbol{D}^{-1/2}$, the Rayleigh quotient $R_{\mathcal{G}}(\boldsymbol{X}) = R_{\mathcal{G}}(f_{\text{Uconv}}(\boldsymbol{X}))$ is invariant under normalized unitary or orthogonal graph convolution (see Definitions 1 and 3).*

In contrast, oversmoothing commonly occurs with vanilla graph convolution and has been proven to occur in a variety of settings [RBM23, CW20, BDGC+22, Ker22]. To illustrate this, we exhibit a simple setting below commonly found at initialization where the parameterized matrix is set to be an orthogonal matrix and input features are random. Here, the magnitude of oversmoothing concentrates around its average and grows with the value of $\operatorname{Tr}(\widetilde{\boldsymbol{A}}^3)$ which corresponds to the (weighted) number of triangles in the graph.

---

[7]This quantity has various equivalent names including the Dirichlet energy, local variance, and Laplacian quadratic form.

**Proposition 7.** *Given a simple undirected graph $\mathcal{G}$ on n nodes with normalized adjacency matrix $\widetilde{A} = D^{-1/2} A D^{-1/2}$ and node degree bounded by D, let $X \in \mathbb{R}^{n \times d}$ have rows drawn i.i.d. from the uniform distribution on the hypersphere in dimension d. Let $f_{\text{conv}}(X) = \widetilde{A} X W$ denote convolution with orthogonal feature transformation matrix $W \in O(d)$. Then, the event below holds with probability $1 - \exp(-\Omega(\sqrt{n}))$:*

$$R_{\mathcal{G}}(X) \geq 1 - O\left(\frac{1}{n^{1/4}}\right) \quad and \quad R_{\mathcal{G}}(f_{\text{conv}}(X)) \leq 1 - \frac{\text{Tr}(\widetilde{A}^3)}{\text{Tr}(\widetilde{A}^2)} + O\left(\frac{1}{n^{1/4}}\right). \tag{15}$$

**Vanishing/Exploding gradients** A commonly observed issue in training deep neural networks, especially RNNs with evolving hidden states, is that gradients can exponentially grow or vanish with depth [HS97, GBC16]. In fact, one of the original motivations for using unitary matrices in RNNs is to directly avoid this vanishing/exploding gradient problem [ASB16, LCMR19, JSD+17, HSL16]. From a theoretical view, prior work has shown that carefully initialized layers have Jacobians that meet variants of the dynamical isometry property commonly studied in the mean field theory literature characterizing the growth/decay over layers of the network [SMG13, XBSD+18, PSG18]. We analyze a version of this property here and discuss it in the context of our work.

**Definition 8** (Dynamical isometry)**.** Given functions $f_1, \ldots, f_L : \mathbb{R}^n \to \mathbb{R}^n$, let $F_i = f_i \circ \cdots \circ f_1$. Let $J_{F_i}(x)$ be the Jacobian matrix of $F_i$ at $x \in \mathbb{R}^{n\,[8]}$. The function $F_L = f_L \circ \cdots \circ f_1$ is dynamically isometric up to $\epsilon$ at $x \in \mathbb{R}^n$ if there exists orthogonal matrix $V \in O(n)$ such that $\left\| \prod_{i=1}^{L} J_{F_i}(x) - V \right\| \leq \epsilon$ where $\| \cdot \|$ denotes the operator norm.

Network layers that meet the dynamical isometry property generally avoid vanishing/exploding gradients. The particular form we analyze is stricter than those studied in the mean field and Gaussian process literature which analyze the distribution of singular values over the randomness of the weights [PSG18, XBSD+18]. Unitary convolution layers followed by isometric activations are examples of dynamical isometries that hold throughout training as exemplified below.

**Example 2.** Compositions of the layer GroupSort($f_{\text{Uconv}}(x)$) consisting of the unitary convolution layer (Definition 1) followed by the Group Sort activation (Eq. (61)) are perfectly dynamically isometric ($\epsilon = 0$) at all $x \in \mathbb{C}^{n\,[9]}$.

## 5 Experiments

Our experimental results here show that unitary/orthogonal variants of graph convolutional networks perform competitively on various graph learning tasks. Due to space constraints, we present experiments on additional datasets and architectures in App. F. This includes experiments on TU-Dataset [MKB+20] and an instance of unitary group convolutional networks on the dihedral group where the goal is to learn distances between pairs of elements in the dihedral group. Training procedures and hyperparameters are reported in App. G. Reported results in tables are over the mean plus/minus standard deviation.

**Toy model: graph distance** To analyze the ability of our unitary GNN to learn long-range dependencies, we consider a toy dataset where the aim is to learn the distance between two indicated nodes on a large graph connected as a ring. This task is inspired by similar toy models on ring graphs where prior work has shown that message passing architectures fail to learn long-range dependencies between distant nodes [DGGB+23]. The particular dataset we analyze consists of a training set of $N = 1000$ graphs on $n = 100$ nodes where each graph is connected as a ring (see Fig. 2a). Node features $x_i \in \mathbb{R}$ are a single number set to zero for all but two randomly chosen nodes whose features are set to one. The goal is to predict the distance between these two randomly chosen nodes. For a graph of $n$ nodes, conventional message passing architectures require at least $n/2$ sequential messages to fully learn this dataset. As shown in Fig. 2b, conventional message passing networks fail to learn this task whereas the unitary convolutional architecture succeeds. We refer the reader to App. F.1 for further details and results for additional architectures.

---

[8]For complex valued inputs, we additionally require that the functions are holomorphic. For sake of simplicity, we will treat functions $f : \mathbb{C}^n \to \mathbb{C}^n$ as real-valued functions $f : \mathbb{R}^{2n} \to \mathbb{R}^{2n}$.

[9]Technically, this holds for almost all $x \in \mathbb{C}^n$ due to non-differentiability of the activation at singular points, but this is handled in practice by choosing a gradient in the sub-differential which meets the criteria.

| | METHOD | PEPTIDES-FUNC Test AP ↑ | PEPTIDES-STRUCT Test MAE ↓ | COCO Test F1 ↑ | PASCAL-VOC Test F1 ↑ |
|---|---|---|---|---|---|
| MP | GCN [†] | 0.6860 ± 0.0050 | 0.2460 ± 0.0007 | 0.1338 ± 0.0007 | 0.2078 ± 0.0031 |
| | GINE [†] | 0.6621 ± 0.0067 | 0.2473 ± 0.0017 | 0.2125 ± 0.0009 | 0.2718 ± 0.0054 |
| | GATEDGCN [†] | 0.6765 ± 0.0047 | 0.2477 ± 0.0009 | 0.2922 ± 0.0018 | 0.3880 ± 0.0040 |
| | GUMP | 0.6843 ± 0.0037 | 0.2564 ± 0.0023 | - | - |
| OTHERS | GPS [†] | 0.6534 ± 0.0091 | 0.2509 ± 0.0014 | **0.3884 ± 0.0055** | 0.4440 ± 0.0065 |
| | DREW | 0.7150 ± 0.0044 | 0.2536 ± 0.0015 | - | 0.3314 ± 0.0024 |
| | EXPHORMER | 0.6527 ± 0.0043 | 0.2481 ± 0.0007 | 0.3430 ± 0.0008 | 0.3960 ± 0.0027 |
| | GRIT | 0.6988 ± 0.0082 | 0.2460 ± 0.0012 | - | - |
| | GRAPH VIT | 0.6942 ± 0.0075 | 0.2449 ± 0.0016 | - | - |
| | CRAWL | 0.7074 ± 0.0032 | 0.2506 ± 0.0022 | - | **0.4588 ± 0.0079** |
| OURS | UNIGCN | 0.7072 ± 0.0035 | **0.2425 ± 0.0009** | 0.2852 ± 0.0016 | 0.3516 ± 0.0070 |
| | LIE UNIGCN | **0.7173 ± 0.0061** | 0.2460 ± 0.0011 | 0.3153 ± 0.0035 | 0.4005 ± 0.0067 |

[†] REPORTED PERFORMANCE TAKEN FROM [TRRG23].

Table 1: Unitary GCN with UniConv (Definition 1) and Lie UniConv (Definition 3) layers compared with other GNN architectures on LRGB datasets [DRG+22]. Top performer bolded and second/third underlined. Networks are set to fit within a parameter budget of 500,000 parameters. Complex numbers are counted as two parameters each. See App. G for additional details.

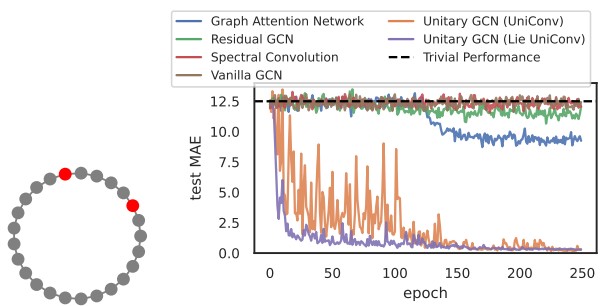

(a) Data sample   (b) Results for message passing architectures

Figure 2: (a) Example datapoint on $n = 25$ nodes; the target is $y = 5$ (distance between red nodes). (b) Results for the ring toy model problem with 100 nodes where the unitary GCN with UniConv or Lie UniConv layers is the only message passing architecture able to learn successfully. Best performance over networks with 5, 10, and 20 layers is plotted. Other architectures typically perform best with 5 layers and only learn shorter distances (see App. F.1).

**Long Range Graph Benchmark (LRGB)**   We consider the Peptides, Coco, and Pascal datatsets from the Long Range Graph Benchmark (LRGB) [DRG+22]. There are two tasks associated with Peptides, a peptide function classification task (Peptides-func) and a regression task (Peptides-struct). Coco and Pascal are node classification tasks. Table 1 shows that the Unitary GCN outperforms standard message passing architectures and is competitive with other state of the art architectures as well, many of which employ global attention mechanisms. This provides evidence that the unitary GCN is nearly as effective at learning long-range signals.

**Heterophilous Graph Dataset**   For node classification, we consider the Heterophilous Graph Dataset proposed by [PKD+23]. The dataset contains the heterophilous graphs Roman-empire, Amazon-ratings, Minesweeper, Tolokers, and Questions, which are often considered as a benchmark for evaluating the performance of GNNs on graphs where connected nodes have dissimilar labels and features. Our results in Table 2 show that the unitary GCN outperforms the baseline message passing and graph transformer models. Given the heterophilous nature of this dataset, these findings reinforce the notion that unitary convolution enhances the ability of convolutional networks to capture long-range dependencies.

# 6 Discussion

In this paper we introduced unitary graph convolutions for enhancing stability in graph neural networks. We provided theoretical and empirical evidence for the effectiveness of our approach. We further introduce an extension to general groups (generalized unitary convolutions), which can be leveraged in group-convolutional architectures to enhance stability.

| METHOD | ROMAN-E. Test AP ↑ | AMAZON-R. Test AP ↑ | MINESWEEPER ROC AUC ↑ | TOLOKERS ROC AUC ↑ | QUESTIONS ROC AUC ↑ |
|---|---|---|---|---|---|
| MP — GCN † | 73.69 ± 0.74 | 48.70 ± 0.63 | 89.75 ± 0.52 | 83.64 ± 0.67 | 76.09 ± 1.27 |
| MP — SAGE † | 85.74 ± 0.67 | 53.63 ± 0.39 | 93.51 ± 0.57 | 82.43 ± 0.44 | 76.44 ± 0.62 |
| MP — GAT † | 80.87 ± 0.30 | 49.09 ± 0.63 | 92.01 ± 0.68 | 83.70 ± 0.47 | 77.43 ± 1.20 |
| MP — GT † | 86.51 ± 0.73 | 51.17 ± 0.66 | 91.85 ± 0.76 | 83.23 ± 0.64 | 77.95 ± 0.68 |
| Ours — UNITARY GCN | **87.21 ± 0.76** | **55.34 ± 0.74** | 94.27 ± 0.58 | 84.83 ± 0.68 | **79.21 ± 0.79** |
| Ours — LIE UNITARY GCN | 85.50 ± 0.22 | 52.35 ± 0.26 | **96.11 ± 0.10** | **85.18 ± 0.43** | 80.01 ± 0.43 |

† REPORTED PERFORMANCE TAKEN FROM [PKD+23].

Table 2: Comparison of Unitary GCN with UniConv (Definition 1) and Lie UniConv (Definition 3) layers with other GNN architectures on the Heterophilous Graph Datasets.

**Limitations** Perhaps the biggest challenge in working with unitary convolutions is the overhead associated with maintaining unitarity or orthogonality via approximations of the exponential map or diagonalizations in Fourier or spectral bases. For implementing unitary maps, working with complex numbers also requires different initialization and activation functions. We refer the reader to App. C.2 and E for methods to alleviate these challenges in practice. Separately, there may be target functions or problem instances where unitarity or orthogonality may not be appropriate. For example, one can envision node classification tasks where the target function is neither (approximately) invertible nor isometric. In such instances, non-unitary layers will be required to learn the task. More generally, deciding when to use unitary layers is problem-dependent. In some cases, such as applications with smaller input graphs, simple and more efficient interventions such as adding residual connections or including batch norm will likely suffice for addressing signal propagation problems.

**Future Directions** In the graph domain, extensions of graph convolution to more advanced methods such as those that better incorporate edge features could widen the range of applications. Exploring hybrid models that combine unitary and non-unitary layers (e.g. global attention mechanisms) could potentially lead to more robust and versatile graph neural networks. Future work can also improve the efficiency of the parameterizations and implementations of the exponential map (see App. C.2). In a similar vein, it is likely that approximately unitary/orthogonal layers suffice in many settings to achieve the performance gains we see in our work. Methods that approximately enforce or regularize layers towards unitarity may be of interest in these instances due to their potential for improved efficiency. In this study, we mainly focused on applications to graph classification and regression tasks; however, the proposed methodology is much more general and could open up a wider range of applications to domains with more general symmetries or different data domains. For example, unitary matrices offer provable guarantees to adversarial attacks (see Corollary 9) and testing this robustness in practice on geometric data has yet to be conducted.

## Acknowledgements

We thank Derek Lim and Andrew Cheng for insightful discussions and Stephen Becker for finding an error in the definition of matrix Lie groups. BK and MW were supported by the Harvard Data Science Initiative Competitive Research Fund and NSF award 2112085.

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

## Table of Contents

## A   Extended related works

### A.1   Further related literature

**Unitary RNNs**   Unitary neural networks were initially developed to tackle the challenge of vanishing and exploding gradients encountered in recurrent neural networks (RNNs) processing lengthy sequences of data. Aiming for more efficient information learning compared to established non-unitary architectures like the long-short term memory unit (LSTM) [HS97], early approaches ensured unitarity through a sequence of parameterized unitary transformations [ASB16, MHRB17, JSD+17]. Other methods like the unitary RNN (uRNN) [WPH+16], the Cayley parameterization (scoRNN) [HWY18], and exponential RNN (expRNN) [LCMR19] parameterized the entire unitary space, maintaining unitarity via Cayley transformations or parameterizing the Lie algebra of the unitary group and performing the exponential map. To eliminate the reliance on the matrix inversion or matrix exponential in these prior algorithms which are computationally expensive in higher dimensions, [KBLL22] showed how to fully parameterize unitary operations more efficiently when applying gradient updates in a low rank subspace.

**Group convolutional neural networks**   For convolutional neural networks acting on data in lattices or grids (e.g. images), various algorithms have proposed orthogonal or unitary versions of linear convolutions over the cyclic group [LHA+19, SF21, TK21, KBLL22]. [XBSD+18] study the task of initializing convolution layers to be an isometry and thereby are able to train very deep CNNs with thousands of layers without the need for residual connections or batch norm. [SGL18] project convolutions onto an operator-norm ball to ensure the norm of the convolution is within a given range. [LHA+19] introduced a block convolutional orthogonal parameterization (BCOP) parameterizing a

subspace of orthogonal convolution operations. [SF21] implement orthogonal convolutions by parameterizing the Lie algebra of the orthogonal group and approximating the exponential map. Their approach is a special case of the instances we discuss in our work. [TK21] and [KBLL22] perform convolution in the Fourier domain where convolution is block diagonalized and unitarity can be enforced across each of these blocks. Generally, unitary convolutional architectures do not perform as well as their vanilla counterparts in terms of accuracy on large image datasets such as CIFAR or ImageNet [TK21, SF21, LJW+19]. Nonetheless, enforcing unitarity here can provide other benefits such as improved robustness to adversarial attacks or added stability with many layers [TK21, KBLL22, SF21, XBSD+18]. Various architectures for more general geometric domains have been proposed [KT18, CGKW18, CW16, GBH18, WZR+20, FSIW20], including generalized group convolutional neural networks considered here. To the best of our knowledge, no explicit extensions of unitary convolutions and related stability aspects have been considered in these settings.

**GNNs and oversmoothing**   Our GNN architectures develop on seminal work proposing variants of graph convolution architectures [KW16, BL17]. Graph convolution is one of various types of trainable layers that act on graphs and are equivariant to the permutation group; among those we compare to in our experiments are [KW16, BL17, XHLJ18, TRRG23, HHL+23, GDBDG23, RGD+22, MLL+23, SVV+23, TRWG23, HYL17, VCC+17, SHF+20]. Recently, some work has proposed variants of unitary/orthogonal graph convolution which we discuss in more detail in the next section (App. A.2). Compared to our method, these either only enforce unitarity in part of the transformation or are expensive to compute. For example, [GZH+22, AEL+24] propose variants of graph convolution which enforce orthogonality in the feature transformation but not in the message passing update itself. [QBY24] propose a graph unitary message passing algorithm that is in general close to an isometry, but scales poorly with the graph size (see App. A.2). Some recent work has also proposed performing message passing in the complex domain. [MPBK24] show a continuous time message passing update that can be made unitary as discussed in App. A.2. [BSC+24] propose an equivariant matrix function graph neural network layer that performs a global update by taking a trainable pole expansion of a matrix function of the adjacency matrix. This layer is effective at incorporating global information, but requires matrix inversion operations that can scale poorly with dimension. [MBB24] give a so-called "balanced orthogonal" initialization for message passing networks using graph attention (GAT). The weight matrix parameters in the GAT layer are initialized as some orthogonal matrix which improves stability for GNNs of the given form with many layers. Nonetheless, the layer itself does not implement an isometry due to the presence of the attention mechanism in the GAT layer.

Various architectures have been proposed to avoid oversmoothing and incorporate nonlocal graph information more generally. This includes approaches that perform small perturbations of the input graph (rewiring [RBM23, FW24]), as well as architectural interventions, such as skip connections [PTPV17, HYL17, XLT+18], which can improve stability in homophilous settings.

**Other settings**   In the context of Bayesian statistics and graphical algorithms, there are variants of the approximate message passing algorithm which leverage properties of unitarity or orthogonality in the message passing [LGJ+21, GX15, MP17]. Unitary equivariant operators are also used in the context of quantum computation and quantum variational algorithms [NSB+22, CNDP+21, SCY+23, SLN+24]. These papers generally study a very different context to that of classical machine learning algorithms. Furthermore, it is unclear whether practical instances of these algorithms offer improvements in comparison to classical machine learning algorithms [AK22, ABKL23]. Separate from quantum machine learning, the unitary graph convolution used here is partly inspired by quantum walks which feature similar unitarity properties [Kem03].

## A.2   Comparison with other proposals for unitary message passing

We compare here previous GNN architectures which have proposed variants of complex-valued or isometric message passing. To facilitate this comparison, first let us recall some standard notation and implementations. Given an (possibly normalized) adjacency matrix $A \in \mathbb{R}^{n \times n}$ acting on $n$ nodes and input features $\mathbf{X} \in \mathbb{R}^{n \times d}$ of dimension $d$, a basic linear message passing layer $f : \mathbb{R}^{n \times d} \to \mathbb{R}^{n \times d'}$ takes the form

$$f_W(\mathbf{X}) = \sigma(A\mathbf{X}W), \tag{16}$$

where $\sigma : \mathbb{R} \rightarrow \mathbb{R}$ is a pointwise nonlinearity and $W \in \mathbb{R}^{d \times d'}$ is a parameterized weight matrix transforming the node features. There are variations of the above which normalize the adjacency matrix or add residual connections, but for sake of comparison, we will study the simplified form above. In contrast, the unitary message passing layer we propose takes the form

$$f_W^{\text{uni}}(\mathbf{X}) = \sigma(\exp(itA)\mathbf{X}W), \tag{17}$$

where $\mathbf{X}$ and $W$ are now complex-valued and $t \in \mathbb{R}$ is a parameter controlling the strength of message passing.

[GZH$^+$22] propose a version of an orthogonal GNN which enforces either exactly or approximately that the matrix $W$ in Eq. (16) is orthogonal, i.e.

$$f_W(\mathbf{X}) = \sigma(A\mathbf{X}W), \text{ where } W^\top W = I. \tag{18}$$

They also discuss regularization of node transformations $W$ to bias towards orthogonality. [AEL$^+$24] also detail a version of a GNN which enforces orthogonality in the matrix $W$. We note that our study focuses on orthogonality in the node evolution during message passing. We discuss in the main text how to combine this with orthogonality in the feature transformation $W$.

Our work also has overlap with the recently proposed fractional Graph Laplacian approach of [MPBK24]. There, they analyze a continuous-time message passing scheme called the fractional Schrödinger equation taking the form

$$\frac{\partial \mathbf{X}}{\partial t} = iL^\alpha \mathbf{X}W, \tag{19}$$

where $\alpha > 0$ is a chosen parameter and $L$ is the normalized graph Laplacian. Unitarity can be strictly enforced in this continuous time format although [MPBK24] do not take this approach. In fact, solving this continuous-time differential equation for some time $t$, we obtain $\text{vec}(\mathbf{X})(t) = \exp(itL^\alpha \otimes W) \text{vec}(\mathbf{X})(0)$ where $\text{vec}(\mathbf{X})(t)$ is the vectorized version of $\mathbf{X}$ at time $t$. Constraining $L^\alpha \otimes W$ to be Hermitian obtains a unitary transformation. Beyond enforcing unitarity, our method works over discrete time, and we find it more efficient to parameterize the weight matrix $W$ outside of the exponential map.

A recent preprint proposes a unitary message passing algorithm called *graph unitary message passing* (GUMP) which sends messages through a larger ring graph constructed from the edges of the original graph [QBY24]. Given a graph $G$ with nodes and edges $V$ and $E$ respectively, the steps of this approach loosely are as follows:

1. Construct a new directed graph (digraph) $G'$ with the same nodes and include directed edges $(i, j)$ and $(j, i)$ for any undirected edge $(i, j)$ in original graph.

2. Convert $G'$ to its line graph $L(G')$: here, each node is an edge in $G'$ and two nodes in $L(G')$ share an edge if there is a path through the original edges in $G'$ (e.g. nodes corresponding to edges $(i, j)$ and $(j, k)$).

3. Construct new node representations in the line graph where for each edge $(i, j)$ we append the node representations $[\boldsymbol{x}_i, \boldsymbol{x}_j]$.

4. Given the adjacency matrix $A_L$ for $L(G')$, find permutation matrices $P_1, P_2$ which form a block diagonal matrix $P_1^\top A_L P_2$.

5. Project each block to its closest unitary in Frobenius norm according to the closed form projection operation [Kel75]:

$$\widetilde{A}_L = \underset{U \in \mathcal{U}(|L(G')|)}{\arg\min} \|A_L - U\|_F^2 = A_L(A_L^\dagger A_L)^{-\frac{1}{2}}. \tag{20}$$

In the above, $\mathcal{U}(n)$ denotes the set of unitary matrices in $\mathbb{C}^{n \times n}$.

6. Invert the permutations obtaining $P_1 \widetilde{A}_L P_2^\top$.

7. Perform GNN convolution using the adjacency matrix constructed previously for $k$ layers.

8. Attach node representations of $L(G')$ to corresponding nodes of $G$ and append these to the initial representations $\boldsymbol{x}$ to output the final representations.

As evident above, the approach here is rather meticulous and many details of the approach are left out. We refer the reader to the preprint [QBY24] for those details. Key to this procedure is that unitarity is roughly enforced by constructing the matrix $\widetilde{A}_L$ which is itself unitary in the vector space of the line graph $L(G')$. Nonetheless, the map from node representations on the original graph $G$ to new node representations on $G$ will only be approximately unitary as the additional steps here will not guarantee properties of isometry and invertibility in general.

For sake of comparison, we should note that the complexity of this approach (in number of nodes $|V|$ and edges $|E|$) is at least $\omega(|E|^2)$ and in practice $O(|E|^3)$ where the most expensive step is the unitary projection which requires a matrix inverse square root. More efficient parameterizations of this procedure or approximations to the projection exist as noted in prior work [KBLL22, LCMR19, LFT20], but this would only improve runtimes to $\omega(|E|^2)$ at best with additional overhead. Furthermore, the GUMP algorithm expands the memory needed to store hidden states from $O(|V|)$ to $O(|E|)$. In comparison, our approach can be made strictly unitary and is more efficient scaling only a constant factor times standard graph convolution runtime to $O(T|V||E|)$ where $T$ is the truncation in the matrix exponential approximation (see App. C.2) with no change to the way hidden states are stored. Our approach is also noticeably simpler as it only reparameterizes the message passing step to be an exponential map over standard message passing procedures.

# B  Deferred proofs

First, we review the properties shown in Fact 1 that unitary convolution meets the properties of invertibility, isometry, and equivariance. This fact follows virtually immediately from the definition of unitarity and equivariance.

*Proof of Fact 1.* The proofs of isometry and invertibility follow directly from the definition of unitarity. What remains to be shown is the equivariance property. For unitary convolution built using Algorithm 1, equivariance can be checked by noting that the exponential map is a composition of equivariant linear operators. For unitary convolution in the Fourier domain (Algorithm 3), equivariance follows from convolution theorems where linear operators appropriately applied in the block diagonal Fourier basis are equivariant [FH13, KT18]. □

As an application, unitary transformations are often used to provide provable robustness guarantees to adversarial perturbations of inputs [TK21, LHA+19, MMS+17, SGL18].

**Corollary 9** (Certified robustness to adversarial perturbations [TK21]). *A simple consequence of the isometry property of the unitary convolution is that $\|f_{\text{Uconv}}(x) - f_{\text{Uconv}}(y)\| = \|x - y\|$ so the Lipschitz constant of this function is 1. Assume neural network classifier $f : \mathbb{R}^n \to \mathbb{R}^m$ is composed of unitary transformations and 1-Lipschitz bounded nonlinearities followed by an L-Lipschitz transformation and for given input $x$, it has margin $\mathcal{M}_f(x)$ defined as*

$$\mathcal{M}_f(x) = \max_{t \in [m]} \left\{ 0, [f(x)]_t - \max_{i \in [m], i \neq t} [f(x)]_i \right\}. \tag{21}$$

*Then, $f$ is certifiably robust (i.e. classification is unchanged) to perturbations $x + \Delta$ of magnitude $\|\Delta\| < \mathcal{M}_f(x)(\sqrt{2}L)^{-1}$.*

In the main text, we noted in Proposition 4 that the above facts in some way cannot be obtained using the standard graph convolution. We restate this here and prove it.

**Proposition 4.** *Let $f_{\text{conv}} : \mathbb{R}^{n \times d} \to \mathbb{R}^{n \times d}$ be a graph convolution layer of the form*

$$f_{\text{conv}}(X, A) = X W_0 + A X W_1, \tag{12}$$

*where $W_0, W_1 \in \mathbb{R}^{d \times d}$ are parameterized matrices. The linear map $f(\cdot, A) : \mathbb{R}^{n \times d} \to \mathbb{R}^{n \times d}$ is orthogonal for all adjacency matrices $A$ of undirected graphs only if $W_1 = 0$ and $W_0 \in O(d)$ is orthogonal. Furthermore, denoting $J_A \in \mathbb{R}^{nd \times nd}$ as the Jacobian matrix of the map $f_{\text{conv}}(\cdot, A)$, for any choice of $W_0, W_1$, there always exists a normalized adjacency matrix $\hat{A}$ such that*

$$\left\| J_{\hat{A}}^\top J_{\hat{A}} - I \right\| \geq \frac{\|W_1\|_F^2}{2d}, \tag{13}$$

*where $\|M\|$ is the operator norm of matrix $M$.*

*Proof.* Note that for (a potentially normalized) adjacency matrix $A$, the Jacobian $J_A$ in the basis of $\text{vec}(X)$ is equal to

$$J_A = I \otimes W_0^\top + A \otimes W_1^\top. \tag{22}$$

Thus, for the map to be orthogonal, it must hold that

$$I = J_A J_A^\top = I \otimes W_0^\top W_0 + A \otimes W_0^\top W_1 + A \otimes W_1^\top W_0 + A^2 \otimes W_1^\top W_1. \tag{23}$$

First, let $A_0$ be the empty graph on two nodes so $A = 0$, then

$$J_{A_0} J_{A_0}^\top = I \otimes W_0^\top W_0, \tag{24}$$

which means $W_0$ must be orthogonal if $J_{A_0}$ is orthogonal. A simple instance with the desired structure can be constructed as follows: Consider the graph on two nodes, which is connected with adjacency matrix $A_1$. Here, we have

$$J_{A_1} J_{A_1}^\top = I \otimes \left( W_0^\top W_0 + W_1^\top W_1 \right) + A \otimes \left( W_1^\top W_0 + W_0^\top W_1 \right). \tag{25}$$

Note that if $W_0$ is orthogonal, then $W_1 = 0$ if $J_{A_1} J_{A_1}^\top = I$. This proves the first part of the proposition.

For the second part, we take traces and note that

$$\begin{aligned} \text{Tr}\left( J_{A_0} J_{A_0}^\top \right) &= 2\|W_0\|_F^2, \\ \text{Tr}\left( J_{A_1} J_{A_1}^\top \right) &= 2\|W_0\|_F^2 + 2\|W_1\|_F^2. \end{aligned} \tag{26}$$

In contrast, $\text{Tr}(I) = 2d$. Therefore, for any choice of the values of $\|W_0\|_F^2, \|W_1\|_F^2$, it must hold that either

$$\left| \text{Tr}\left( J_{A_0} J_{A_0}^\top \right) - 2d \right| \geq \|W_1\|_F^2 \tag{27}$$

or

$$\left| \text{Tr}\left( J_{A_1} J_{A_1}^\top \right) - 2d \right| \geq \|W_1\|_F^2. \tag{28}$$

W.l.o.g. assume the first event holds. Denoting the singular values of a matrix $J_{A_1} J_{A_1} - I$ as $s_1, \ldots, s_{2d}$, we have

$$\|W_1\|_F^2 \leq \left| \text{Tr}\left( J_{A_1} J_{A_1}^\top - I \right) \right| \leq s_1 + \cdots + s_{2d} \leq 2d \max_i s_i. \tag{29}$$

Rearranging, we obtain the final result. $\qquad\square$

One consequence of the above fact is that fully parameterized unitary graph convolution requires higher order powers of $A$ to be implemented. This is essentially one reason why the exponential map (or some approximation thereof) is needed.

We now show that the Rayleigh quotient defined in Definition 5 is invariant under unitary transformations. As before, this will follow virtually directly from the unitarity properties. We restate the proposition below.

**Proposition 6** (Invariance of Rayleigh quotient). *Given an undirected graph $\mathcal{G}$ on n nodes with normalized adjacency matrix $\widetilde{A} = D^{-1/2} A D^{-1/2}$, the Rayleigh quotient $R_{\mathcal{G}}(X) = R_{\mathcal{G}}(f_{\text{Uconv}}(X))$ is invariant under normalized unitary or orthogonal graph convolution (see Definitions 1 and 3).*

*Proof.* First, consider separable unitary convolution (Definition 1). The Rayleigh quotient takes the form

$$R_{\mathcal{G}}(f_{\text{Uconv}}(X)) = \frac{\text{Tr}\left( \left( \exp(i\widetilde{A}) X U \right)^\dagger (I - \widetilde{A}) \exp(i\widetilde{A}) X U \right)}{\| \exp(i\widetilde{A}) X U \|_F^2}. \tag{30}$$

The Frobenius norm is invariant under unitary transformations so the denominator $\| \exp(i\widetilde{A}) X U \|_F^2 = \|X\|_F^2$. For the numerator we have by the cyclic property of trace

$$\text{Tr}\left( \left( \exp(i\widetilde{A}) X U \right)^\dagger (I - \widetilde{A}) \exp(i\widetilde{A}) X U \right) = \text{Tr}\left( X^\dagger \exp(-i\widetilde{A})(I - \widetilde{A}) \exp(i\widetilde{A}) X \right) \tag{31}$$

The matrices $\exp(-i\widetilde{A}), (I - \widetilde{A}), \exp(i\widetilde{A})$ all share the same eigenbasis so they commute and thus

$$\operatorname{Tr}\left(X^{\dagger}\exp(-i\widetilde{A})(I - \widetilde{A})\exp(i\widetilde{A})X\right) = \operatorname{Tr}\left(X^{\dagger}(I - \widetilde{A})X\right). \tag{32}$$

Thus, we have

$$R_{\mathcal{G}}(f_{\text{Uconv}}(X)) = \frac{\operatorname{Tr}\left(X^{\dagger}(I - \widetilde{A})X\right)}{\|X\|_F^2} = R_{\mathcal{G}}(X). \tag{33}$$

Similarly for Lie unitary/orthogonal convolution (Definition 3), the isometry property guarantees $\|X\|_F^2 = \|f_{\text{Uconv}}(X)\|_F^2$. Furthermore, when viewed as a linear map in the basis of $\operatorname{vec}(X) \in \mathbb{C}^{nd}$ (i.e. entries of $X$ viewed as a vector), $f_{\text{Uconv}}$ can be written as a linear map of the form

$$\operatorname{vec}(f_{\text{Uconv}}(X)) = \exp(A \otimes W^{\top})\operatorname{vec}(X), \tag{34}$$

where $W$ is the feature transformation matrix in Definition 3. Finally, note that $\exp(A \otimes W^{\top})$ commutes with $A$ so

$$\begin{aligned}
&\operatorname{Tr}\left(f_{\text{Uconv}}(X)^{\dagger}(I - \widetilde{A})f_{\text{Uconv}}(X)\right) \\
&= \operatorname{vec}(X)^{\dagger}\exp(A \otimes W^{\top})^{\dagger}\left[(I - \widetilde{A}) \otimes I\right]\exp(A \otimes W^{\top})\operatorname{vec}(X) \\
&= \operatorname{vec}(X)^{\dagger}\left[(I - \widetilde{A}) \otimes I\right]\operatorname{vec}(X).
\end{aligned} \tag{35}$$

Multiplying the above by $\|X\|_F^{-2}$ recovers $R_{\mathcal{G}}(X)$. $\qquad\square$

In contrast, we gave an example (Proposition 7) where the Rayleigh quotient decays with high probability for vanilla convolution, which we restate below.

**Proposition 7.** *Given a simple undirected graph $\mathcal{G}$ on $n$ nodes with normalized adjacency matrix $\widetilde{A} = D^{-1/2}AD^{-1/2}$ and node degree bounded by $D$, let $X \in \mathbb{R}^{n \times d}$ have rows drawn i.i.d. from the uniform distribution on the hypersphere in dimension $d$. Let $f_{\text{conv}}(X) = \widetilde{A}XW$ denote convolution with orthogonal feature transformation matrix $W \in O(d)$. Then, the event below holds with probability $1 - \exp(-\Omega(\sqrt{n}))$:*

$$R_{\mathcal{G}}(X) \geq 1 - O\left(\frac{1}{n^{1/4}}\right) \quad\text{and}\quad R_{\mathcal{G}}(f_{\text{conv}}(X)) \leq 1 - \frac{\operatorname{Tr}(\widetilde{A}^3)}{\operatorname{Tr}(\widetilde{A}^2)} + O\left(\frac{1}{n^{1/4}}\right). \tag{15}$$

*Proof.* Note that

$$\mathbb{E}\left[R_{\mathcal{G}}(X)\right] = \mathbb{E}\left[\frac{\operatorname{Tr}\left(X^{\top}(I - \widetilde{A})X\right)}{\|X\|_F^2}\right] = \frac{1}{n}\mathbb{E}\left[\operatorname{Tr}\left((I - \widetilde{A})XX^{\top}\right)\right]. \tag{36}$$

By symmetry properties of the uniform distribution $\operatorname{Unif}(S^{d-1})$ on the hypersphere, $\mathbb{E}[XX^{\top}] = I$. Thus,

$$\mathbb{E}\left[R_{\mathcal{G}}(X)\right] = 1. \tag{37}$$

Furthermore, treating $R_{\mathcal{G}}(X)$ as a function of its node features $x_1, \ldots, x_n$ where $x_i = X_{i,:}^{\top}$, we have by the Azuma-Hoeffding inequality that

$$\mathbb{P}\left[\left|R_{\mathcal{G}}(X) - \mathbb{E}R_{\mathcal{G}}(X)\right| \geq \epsilon\right] \leq \exp(-\Omega(n\epsilon^2)). \tag{38}$$

The above can be shown by noting that the operator norm of $I - \widetilde{A}$ is bounded by two and changing the values of any $x_i$ changes $R_{\mathcal{G}}(X)$ by at most $4/n$. Therefore, setting $\epsilon = \varepsilon n^{-1/4}$, we get that with probability $1 - \exp(-\Omega(\varepsilon^2\sqrt{n}))$,

$$R_{\mathcal{G}}(X) = 1 - O(1/n^{1/4}). \tag{39}$$

For $R_{\mathcal{G}}(f_{\text{conv}}(\boldsymbol{X}))$, we have

$$
\begin{aligned}
\mathbb{E}\left[R_{\mathcal{G}}(f_{\text{conv}}(\boldsymbol{X}))\right] &= \mathbb{E}\left[\frac{\text{Tr}\left(\boldsymbol{W}^\top \boldsymbol{X}^\top \widetilde{\boldsymbol{A}}(\boldsymbol{I}-\widetilde{\boldsymbol{A}})\widetilde{\boldsymbol{A}}\boldsymbol{X}\boldsymbol{W}\right)}{\|\widetilde{\boldsymbol{A}}\boldsymbol{X}\boldsymbol{W}\|_F^2}\right] \\
&= \mathbb{E}\left[\frac{\text{Tr}\left(\boldsymbol{X}^\top \widetilde{\boldsymbol{A}}(\boldsymbol{I}-\widetilde{\boldsymbol{A}})\widetilde{\boldsymbol{A}}\boldsymbol{X}\right)}{\|\widetilde{\boldsymbol{A}}\boldsymbol{X}\|_F^2}\right] \\
&= \mathbb{E}\left[1 - \frac{\text{Tr}\left(\boldsymbol{X}^\top \widetilde{\boldsymbol{A}}^3 \boldsymbol{X}\right)}{\|\widetilde{\boldsymbol{A}}\boldsymbol{X}\|_F^2}\right].
\end{aligned}
\tag{40}
$$

Again, using the Azuma-Hoeffding argument as before, the numerator above concentrates around its expectation as

$$
\mathbb{P}\left[\left|\text{Tr}\left(\boldsymbol{X}^\top \widetilde{\boldsymbol{A}}^3 \boldsymbol{X}\right) - \mathbb{E}\,\text{Tr}\left(\boldsymbol{X}^\top \widetilde{\boldsymbol{A}}^3 \boldsymbol{X}\right)\right| \geq \epsilon n\right] \leq \exp(-\Omega(\epsilon^2 n)).
\tag{41}
$$

A similar statement holds for the denominator $\|\widetilde{\boldsymbol{A}}\boldsymbol{X}\|_F^2$. Furthermore, we have by symmetry properties of the distribution of $\boldsymbol{X}$ and linearity of expectation:

$$
\mathbb{E}\,\text{Tr}\left(\boldsymbol{X}^\top \widetilde{\boldsymbol{A}}^3 \boldsymbol{X}\right) = \text{Tr}(\widetilde{\boldsymbol{A}}^3), \quad \mathbb{E}\|\widetilde{\boldsymbol{A}}\boldsymbol{X}\|_F^2 = \text{Tr}(\widetilde{\boldsymbol{A}}^2).
\tag{42}
$$

Combining the above facts and applying the union bound, we have that with probability $1 - \exp(\Omega(\varepsilon^2 \sqrt{n}))$,

$$
1 - \frac{\text{Tr}\left(\boldsymbol{X}^\top \widetilde{\boldsymbol{A}}^3 \boldsymbol{X}\right)}{\|\widetilde{\boldsymbol{A}}\boldsymbol{X}\|} \leq 1 - \frac{\text{Tr}(\widetilde{\boldsymbol{A}}^3) - \varepsilon n^{3/4}}{\text{Tr}(\widetilde{\boldsymbol{A}}^2) + \varepsilon n^{3/4}} = 1 - \frac{\text{Tr}(\widetilde{\boldsymbol{A}}^3)}{\text{Tr}(\widetilde{\boldsymbol{A}}^2)} + O(n^{-1/4}).
\tag{43}
$$

In the last equality, we use the fact that $\boldsymbol{A}$ has bounded degree and thus $\text{Tr}(\widetilde{\boldsymbol{A}}^2) = \Theta(n)$. □

# C  Background on representation theory, Lie groups, exponential map, and related approximations

## C.1  Matrix Lie groups

We give a brief overview of matrix Lie groups and Lie algebras here and recommend [Hal15] for detailed exposition. Matrix Lie groups are subsets of invertible matrices that form a differentiable manifold formally defined below.

**Definition 10** (Matrix Lie groups [Hal15]). A matrix Lie group is any subgroup of $GL(n, \mathbb{C})$ with the property that any convergent sequence of matrices $M_m \in \mathbb{C}^{n \times n}$ in the subgroup converge to a matrix $\boldsymbol{M}$ that is either an element of the subgroup or not invertible (*i.e.*, not in $GL(n, \mathbb{C})$).

The orthogonal and unitary groups whose definitions are copied below meet these criteria.

$$
O(n) = \left\{\boldsymbol{M} \in \mathbb{R}^{n \times n} | \boldsymbol{M}\boldsymbol{M}^\top = \boldsymbol{M}^\top \boldsymbol{M} = \boldsymbol{I}\right\},
\tag{44}
$$

$$
U(n) = \left\{\boldsymbol{M} \in \mathbb{C}^{n \times n} | \boldsymbol{M}\boldsymbol{M}^\dagger = \boldsymbol{M}^\dagger \boldsymbol{M} = \boldsymbol{I}\right\}.
\tag{45}
$$

A crucial operation in matrix Lie groups is the exponential map. Given a matrix $\boldsymbol{M} : C^{n \times n}$ which is an endomorphism $\text{End}(\mathbb{C}^n)$ on the vector space $\mathbb{C}^n$, the exponential map $\exp : \text{End}(\mathbb{C}^n) \to \text{End}(\mathbb{C}^n)$ returns another matrix (endomorphism) and is defined as

$$
\exp(\boldsymbol{M}) = \sum_{p=0}^{\infty} \frac{1}{p!} \boldsymbol{M}^p.
\tag{46}
$$

The exponential map, as we will show below, maps from the Lie algebra to the Lie group. It also has an interpretation over Riemannian manifolds which we do not discuss here. We refer the reader to a

textbook [Pet06] or prior work in machine learning [LCMR19] for further details of that connection. For compact groups, the exponential map is a smooth map whose image is the connected component to the identity of the Lie group [KJ08, Hal15]. The Lie algebra is the tangent space of a Lie group at the identity element. To see this, note that

$$\frac{d}{dt} \exp(t\boldsymbol{X}) = \boldsymbol{X} \exp(t\boldsymbol{X}) = \exp(t\boldsymbol{X})\boldsymbol{X}, \tag{47}$$

and

$$\frac{d}{dt} \exp(t\boldsymbol{X})\bigg|_{t=0} = \boldsymbol{X}. \tag{48}$$

Note that $\exp(\boldsymbol{0}) = \boldsymbol{I}$. The above gives us the Lie algebra to a given group.

**Definition 11** (Lie algebra [Hal15]). Given a matrix Lie group $G$, the Lie algebra $\mathfrak{g}$ of $G$ is the set of matrices $\boldsymbol{X}$ such that $e^{t\boldsymbol{X}} \in G$ for all $t \in \mathbb{R}$.

As an example, consider the unitary group where given a matrix $\boldsymbol{U} \in U(n)$ and $\boldsymbol{X} \in \mathfrak{u}(n)$. Here, we have

$$\frac{d}{dt} \exp(-t\boldsymbol{X})\bigg|_{t=0} = \frac{d}{dt} \exp(t\boldsymbol{X}^\dagger)\bigg|_{t=0} \implies -\boldsymbol{X} = \boldsymbol{X}^\dagger. \tag{49}$$

## C.2 Exponential map

First, let us recall the definition of the exponential map. Given a linear operator $\mathsf{L} : \mathcal{V} \to \mathcal{V}$ which is an endomorphism $\mathrm{End}(\mathcal{V})$ on a vector space $\mathcal{V}$, the exponential map $\exp : \mathrm{End}(\mathcal{V}) \to \mathrm{End}(\mathcal{V})$ is defined as

$$\exp(\mathsf{L})(\mathbf{X}) = \sum_{p=0}^{\infty} \frac{1}{p!} \mathsf{L}^p(\mathbf{X}) = \mathbf{X} + \mathsf{L}(\mathbf{X}) + \frac{1}{2}\mathsf{L} \circ \mathsf{L}(\mathbf{X}) + \frac{1}{6}\mathsf{L} \circ \mathsf{L} \circ \mathsf{L}(\mathbf{X}) + \cdots \tag{50}$$

Applying the exponential map of a linear operator to a given vector in a vector space of dimension $n$ to computer precision can scale in practice as $O(n^3)$ similarly to performing an eigendecomposition. In fact, for matrices $\boldsymbol{M} \in \mathbb{C}^{n \times n}$ which have an eigendecomposition $\boldsymbol{M} = \boldsymbol{U}\boldsymbol{D}\boldsymbol{U}^\dagger$ (for skew-Hermitian matrices $\boldsymbol{M} \in \mathfrak{u}(n)$, the spectral theorem guarantees the existence of an eigendecomposition.), $\exp(\boldsymbol{M}) = \boldsymbol{U}\exp(\boldsymbol{D})\boldsymbol{U}^\dagger$ where $\exp(\boldsymbol{D})$ can be easily implemented by performing the exponential elementwise on the diagonal entries of $\boldsymbol{D}$. However, in practice, we can exploit approximations which avoid expensive operations such as eigendecompositions.

**Taylor approximation**    Often the simplest and most efficient approximation is a $k$-th order truncation to the Taylor series

$$\exp[\mathrm{conv}](\mathbf{X}) \approx \sum_{p=0}^{k} \frac{1}{p!} \mathrm{conv}^{(p)}(\mathbf{X}), \tag{51}$$

where $\mathrm{conv}^p : \mathcal{V} \to \mathcal{V}$ indicates the composition of the operator conv $p$ times. By Taylor's theorem, one can bound the error in this approximation as

$$\left\| \exp[\mathrm{conv}](\mathbf{X}) - \sum_{p=0}^{k} \frac{1}{p!} \mathrm{conv}^p(\mathbf{X}) \right\|_2 \leq O\left( \frac{\|\mathrm{conv}\|^{k+1}\|\mathbf{X}\|_2}{(k+1)!} \right),$$

where $\|\mathrm{conv}\|$ denotes the operator norm of conv. Therefore, error decreases exponentially with $k$. In practice, we find setting $k = 12$ is sufficiently accurate for the GNNs we train in our experiments. Of course, $k$ can be increased in settings with larger graphs or more training time where instabilities are likely to arise. We also by default perform unitary convolution with normalized adjacency matrices $\boldsymbol{D}^{-1/2}\boldsymbol{A}\boldsymbol{D}^{-1/2}$ ($\boldsymbol{D}$ is the diagonal degree matrix) so that the operator norm of the linear convolution operation is bounded by one.

**Padé approximant**  Padé approximations are optional rational functional approximations of a given function up to a given order. In unitary settings, Padé approximants are often preferred because they return a unitary operator. In contrast, Taylor approximations discussed previously only return a linear operator which can be made arbitrarily close to a unitary operator. Padé approximants of order $k$ take the form

$$\exp(M) \approx p_k(M)q_k(M)^{-1}, \tag{52}$$

where $p_k, q_k$ are degree $k$ polynomials. The degree $k$ Padé approximant agrees with the Taylor expansion up to order $2k$. The first order Padé approximant, also known as the Cayley map, has been used in prior neural network implementations [WPH+16, HWY18, TK21] and takes the form

$$\exp(M) \approx \left(I + \frac{1}{2}M\right)\left(I - \frac{1}{2}M\right)^{-1}. \tag{53}$$

One practical drawback of Padé approximants are that they typically require implementations of a matrix inverse or approximations thereof which render this more challenging to implement. In our setting, we found that the Taylor approximation sufficed in accuracy and stability so we did not implement this.

Finally, we should remark that Padé approximants can be used to pre-compute matrix exponentials to computer precision [Hig09, AMH10, AMH09]. [LCMR19] use these techniques to parameterize unitary layers accurately for RNNs. One could pre-compute matrix exponentials of adjacency matrices in our setting to speed up training, though we have not implemented this in our experiments. We leave this for future work.

**Other implementations of exponential map**  We briefly mention here for sake of completeness a different approach to approximating the exponential map based on the Baker-Campbell-Hausdorff formula [Hal15] which states that for matrices $X, Y \in \mathbb{C}^{n \times n}$:

$$\exp(X)\exp(Y) = \exp\left(X + Y + \frac{1}{2}[X, Y] + \frac{1}{12}[X, [X, Y]] - \frac{1}{12}[Y, [X, Y]] + \cdots\right). \tag{54}$$

The Lie-Trotter approximation uses the above fact to implement the matrix exponential of a sum of matrices as a product of matrix exponentials over elements of the sum. This method is often used in high dimensional spaces where the output of the exponential map on each Lie algebra generator is known. The first order expansion takes the form [Tro59, CST+21]

$$\exp(X + Y) = \lim_{n \to \infty}\left[\exp\left(\frac{X}{n}\right)\exp\left(\frac{Y}{n}\right)\right]^n \approx \left[\exp\left(\frac{X}{k}\right)\exp\left(\frac{Y}{k}\right)\right]^k, \tag{55}$$

where $k$ is a positive integer controlling the level of approximation. The above is accurate to order $O(1/k)$, and higher order accurate schemes exist. These methods are commonly used in machine learning, quantum computation, and numerical methods[10] [CST+21, MGL+23, Llo96, Str68, MQ02]. One advantage of this approach is that the approximation applied to an operator in the Lie algebra always returns an element in the Lie group since the approximation is a product of Lie group elements themselves.

## D   Fourier implementation of group convolution

In the main text, we described a generalized procedure to implement unitary convolution with parameterized operators in the Lie algebra of a group. We complement that with a mostly informal discussion on how to implement unitary convolution in the Fourier domain. The algorithms are summarized in Algorithm 3 and Algorithm 2.

Before detailing the algorithm in Algorithm 3, we provide a brief overview of Fourier-based convolutions over arbitrary groups, which generalizes the classical Fourier transform to that over arbitrary groups. We recommend [S+77] for a more complete and rigorous background. Throughout this, we will assume that groups are finite, though this can be generalized to other groups, especially those that are compact.

---

[10]In numerical methods, they fall under the umbrella of splitting methods or Strang splitting.

| **Algorithm 2** Unitary map from Lie algebra | **Algorithm 3** Unitary map in Fourier basis |
|---|---|

**Algorithm 2** Unitary map from Lie algebra

**Input:** equivariant linear operator $L \in \mathbb{C}^n \to \mathbb{C}^n$
**Input:** vector $\mathbf{x} \in \mathbb{C}^n$
1: $\widetilde{L} = \frac{1}{2}(L - L^\dagger)$ (skew symmetrize operator)
2: **return** $\exp(\widetilde{L})(\mathbf{x})$ (or approximation thereof)

**Algorithm 3** Unitary map in Fourier basis

**Input:** Fourier transform $\mathcal{F} : \mathbb{C}^n \to \bigoplus_{i=1}^m \mathbb{C}^{d_i \times d_i}$
**Input:** Unitary operators $\{U_i\}_{i=1}^m$, $U_i \in U(d_i)$
**Input:** vector $\mathbf{x} \in \mathbb{C}^n$
1: $Y = \mathcal{F}(\mathbf{x})$ (apply Fourier transform)
2: $Z = \left[\bigoplus_{i=1}^m U_i\right] Y$ (apply block diagonal unitaries)
3: **return** $\mathcal{F}^{-1}(Z)$ (inverse Fourier transform)

A representation of a group is a map $\rho : G \to \mathbb{F}^{d \times d}$ for a given field $\mathbb{F}$ such that $\rho(g)\rho(g') = \rho(gg')$ (homomorphism). A representation is reducible if there exists an invertible matrix $Q \in \mathbb{F}^{d \times d}$ such that $Q\rho(g)Q^{-1} = \oplus_{i=1}^k \rho_i'(g)$ is a direct sum of at least $k \geq 2$ representations $\rho_1', \ldots, \rho_k'$. A representation is irreducible if such a decomposition does not exist. For any finite group $G$, a system of unique irreps $\rho_1, \ldots, \rho_K$ always exists. It holds that $\sum_i \dim(\rho_i)^2 = |G|$ for any such set of unique irreducible representations [S+77]. Here, the uniqueness is to eliminate redundancies due to the fact that any irrep $\rho$ can be mapped to a new one by an invertible transformation $\rho'(g) = Q\rho(g)Q^{-1}$.

The Fourier transform of a function $f : G \to \mathbb{C}$ with respect to a set of irreducible representations (irreps) $\{\rho_i\}_{i=1}^m$ is given by:

$$\hat{f}(\rho_i) = \sum_{u \in G} f(u)\rho_i(u), \quad i = 1, 2, \ldots, m. \tag{56}$$

For abelian groups these irreps are all one dimensional. The set of $\rho_i(u)$ for the cyclic group for example correspond to the entries of the discrete Fourier transform matrix. For non-abelian groups, there always exists an irrep which is at least of dimension 2.

In Algorithm 3, we denote the Fourier transform $\mathcal{F} : \mathbb{C}^n \to \oplus_{i=1}^m \mathbb{C}^{d_i \times d_i}$ as a map that takes in the inputs to a function $f$ and outputs a direct sum of the Fourier basis of the function $\hat{f}$ over the irreps. Given two functions $f, g : G \to \mathbb{C}$, their convolution outputs another function $(f * g) : G \to \mathbb{C}$ and is equal to

$$(f * g)(u) = \sum_{v \in G} f(uv^{-1})g(v). \tag{57}$$

In the main text, we describe these operations as vector operations over a vector space $\mathbb{C}^n$. Setting $n = |G|$ and taking the so-called regular representation as maps acting on $\mathbb{C}^{|G|}$ can recover the form in the main text. Finally, the Fourier transform of their convolution is the matrix product of their respective Fourier transforms:

$$\widehat{(f * g)}(\rho_i) = \hat{f}(\rho_i)\hat{g}(\rho_i), \tag{58}$$

where $\rho_i{}_{i=1}^m$ are the irreps of $G$. The above assumes that all the functions have input domain over the group. This is not strictly necessary and generalizations exist which map functions on homogeneous spaces to the setting above [KT18].

The general implementation of Fourier convolution is given in Algorithm 3. Here, one employs a Fourier operator which block diagonalizes the input into its irreducible representations or some spectral representation. Then, applying blocks of unitary matrices in this representation and inverting the Fourier transform implements a unitary convolution. The details will depend on the particular form of the Fourier transform and irreducible representations. This method is often preferred when filters are densely supported and efficient implementations of the Fourier transform are obtained. Previous implementations have been designed for CNNs [TK21, KBLL22].

**Example 3** (Convolution on regular representation (Fourier basis)). Continuing the previous example, assume the group $G$ has unitary irreducible representations $\rho_1, \ldots, \rho_m$ (irreps) where $\rho_i : G \to \mathbb{C}^{d_i \times d_i}$. The group Fourier transform maps input function $x : G \to \mathbb{C}$ to its irrep basis as [FH13, KT18]

$$\hat{x}(\rho_i) = \sum_{g \in G} x(g)\rho_i(g), \tag{59}$$

and group convolution is now block diagonal in this basis

$$\widehat{(m \star x)}(\rho) = \hat{x}(\rho)\hat{m}(\rho)^\dagger. \tag{60}$$

Implementing unitary convolution requires that $\hat{m}(\rho)$ is unitary for all irreps $\rho$.

# E   Architectural considerations

Handling complex numbers and enforcing isometry in neural networks requires changes to some standard practice in training neural networks. We summarize some of the important considerations here and refer the reader to surveys and prior works for further details [BQL21, TK21, LCMR19, KBLL22].

**Handling different input and output dimensions**   Unitary and orthogonal transformations are defined on input and output spaces of the same dimension. Maintaining isometry for different input and output dimensions formally requires manipulations of the Stiefel manifold as studied in various prior works [LCMR19, LFT20, NA05, JD15]. When the input dimension is less than that of the output dimension, one simple way to implement semi-unitary or semi-orthogonal convolutions via standard unitary layers is simply to pad the inputs with zeros to match the output dimensionality. We also often will simply use a standard (unconstrained) linear transformation to first embed inputs in the given dimension that is later used for unitary transformations.

**Nonlinearities**   For handling complex numbers, one must typically redefine nonlinearities to handle complex inputs. We find that applying standard nonlinearities separately to the real and imaginary parts works well in practice in line with other works [BQL21]. To enforce isometry as well in the nonlinearity, we use the GroupSort : $\mathbb{R}^2 \to \mathbb{R}^2$ activation [TK21, ALG19], which acts on a pair of numbers as

$$\text{GroupSort}(a, b) = (\max(a, b), \min(a, b)), \tag{61}$$

and is clearly norm-preserving. To apply this nonlinearity to a given layer, we split the channels or feature dimension into two separate parts and apply the nonlinearity across the split.

**Initialization**   Given the constraints on unitary matrices and skew-Hermitian matrices, prior work has proposed various forms of initialization that meet the constraints of these matrices. One strategy that has been effective in prior work and proposed in [HSL16, HWY18] is to initialize in $2 \times 2$ blocks along the diagonal. For skew symmetric matrices, one way of achieving this is to initialize $2 \times 2$ blocks as

$$\begin{bmatrix} 0 & s \\ -s & 0 \end{bmatrix}, \tag{62}$$

where $s \sim \text{Unif}(-\pi, \pi)$ for example [HSL16].

**Directed graphs**   Directed graphs present a challenge because their adjacency matrix is not guaranteed to be symmetric. Given an adjacency matrix $A \in \mathbb{R}^{n \times n}$ of a directed graph, one simple way to proceed is to split the directed graph into its symmetric and non-symmetric parts $A = A_{\text{sym}} + A_{\text{nonsym}}$. Here, we assume that for matrix entry $A_{ij}$, either $A_{ij} = A_{ji}$ or $A_{ij} A_{ji} = 0$ (i.e. one of the transposed entries is zero). Then, we set

$$\left[ A_{\text{sym}} \right]_{ij} = \begin{cases} A_{ij} & \text{if } A_{ij} = A_{ji} \\ 0 & \text{otherwise} \end{cases} \tag{63}$$

and

$$\left[ A_{\text{nonsym}} \right]_{ij} = \begin{cases} A_{ij} & \text{if } A_{ij} \neq A_{ji}, \ A_{ij} \neq 0 \\ -A_{ji} & \text{if } A_{ij} \neq A_{ji}, \ A_{ij} = 0 \\ 0 & \text{otherwise} \end{cases} \tag{64}$$

Finally, one can then perform graph convolution with the skew-Hermitian matrix

$$H = i A_{\text{sym}} + A_{\text{nonsym}}. \tag{65}$$

We do not work with directed graphs in our experiments and have not implemented this in our code.

# F   Additional experiments

## F.1   Additional results on toy model of graph distance

Fig. 3 shows additional results for the toy model considered in the main text. As a reminder, this task is to learn the graph distance between pairs of randomly selected nodes in a ring graph of 100

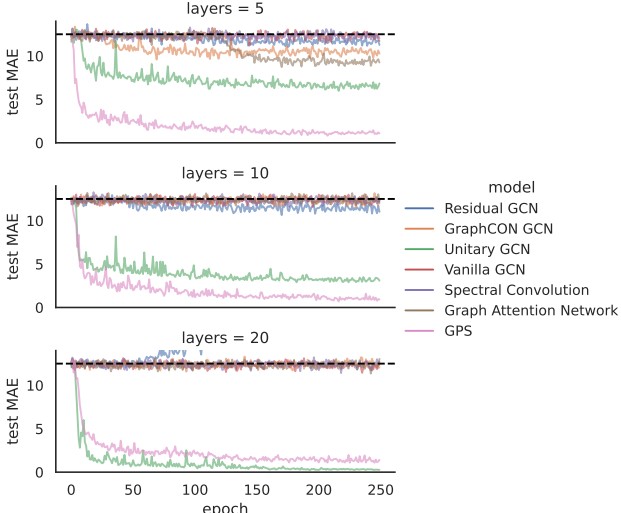

Figure 3: Additional results on the ring plot toy model including additional architectures. We show here the performance of various models with 5, 10, or 20 layers. The unitary GCN is the only message passing architecture that achieves stable performance with added layers and can learn the task. Apart from message passing architectures, global transformer architectures like GPS can learn the task when given Laplacian positional encoding. The trivial performance corresponding to outputting the average output is shown as a dotted horizontal line.

nodes. Message passing architectures need at least 50 sequential messages to fully learn this task. As layers are added to the unitary GCN, the network is able to learn the task better. This is in contrast to other message passing architectures which perform worse with additional layers. Apart from message passing architectures, strong performance is also achieved by transformer architectures like GPS with global attention. The networks we study are the vanilla GCN (Residual GCN includes skip connections) [KW16], graph attention network [VCC+17], spectral convolution [ZK20], transformer-based GPS [RGD+22], and graph-coupled oscillator network (GraphCON), which is a discretization of a second-order ODE on the graph [RCR+22]. Hyperparameters and additional network details are reported in App. G.

## F.2 TU Datasets

We perform experiments on the ENZYMES, IMDB-BINARY, MUTAG, and PROTEINS tasks from the TU Dataset database [MKB+20]. As can be seen in table 3, with the exception of IMDB, a GCN with UniConv layers outperforms all message-passing GNNs tested against on all datasets, by margins of up to 18 percent. We follow the training procedure in [NHN+23] and report results for GCN and GIN from [NHN+23]. For GAT and unitary GCN, we tune the dropout and learning rate as detailed in App. G. All results are accumulated over 100 random trials. In addition, in Fig. 4, we show that the unitary GCN maintains its performance over large network depths in contrast to other message passing layers. The deteriation in performance of conventional message passing layers is a likely a consequence of over-smoothing which we show does not occur in unitary graph convolution [CW20, RBM23].

## F.3 Dihedral group distance

The dihedral group $D_n$ is a group of order $2n$ describing the symmetries (rotations and reflections) of a regular polygon. Its elements are generated by a rotation generator $r$ and reflection generator $s$:

$$D_n = \left\langle r, s \mid r^n = s^2 = (sr)^2 = 1 \right\rangle. \tag{66}$$

In this task, analaogous to the graph distance task in Sec. 5, the goal is to learn the distance between two group elements $g, g'$ in the dihedral group $D_n$. Formally, we aim to learn a target function $f : \mathbb{R}^{|D_n|} \to \mathbb{Z}$ mapping inputs of dimension $2n$ (the vector space of the regular representation) to

| Method | ENZYMES Test AP ↑ | IMDB Test AP ↑ | MUTAG Test AP ↑ | PROTEINS Test AP ↑ |
|---|---|---|---|---|
| **MP** GCN | $25.5 \pm 1.2$ | $49.3 \pm 0.8$ | $68.8 \pm 2.4$ | $70.6 \pm 0.7$ |
| GIN | $31.3 \pm 1.1$ | $\mathbf{69.0 \pm 1.0}$ | $75.7 \pm 3.6$ | $69.7 \pm 0.8$ |
| GAT | $26.8 \pm 1.2$ | $47.0 \pm 1.1$ | $68.5 \pm 2.8$ | $72.6 \pm 0.8$ |
| **Ours** Unitary GCN | $41.5 \pm 1.6$ | $61.2 \pm 1.4$ | $86.8 \pm 3.3$ | $75.1 \pm 1.3$ |
| Wide Unitary GCN | $\mathbf{42.1 \pm 1.5}$ | $62.4 \pm 1.3$ | $\mathbf{87.1 \pm 3.5}$ | $\mathbf{75.6 \pm 1.0}$ |

Table 3: Comparison of Unitary GCN with Lie UniConv layers (Definition 3) with other GNN architectures on the TU datasets. Each complex number is counted as two parameters for our architectures, except for wide Unitary GCN which counts a complex numbers as one parameter so that the width of hidden layers roughly matches that of vanilla GCN.

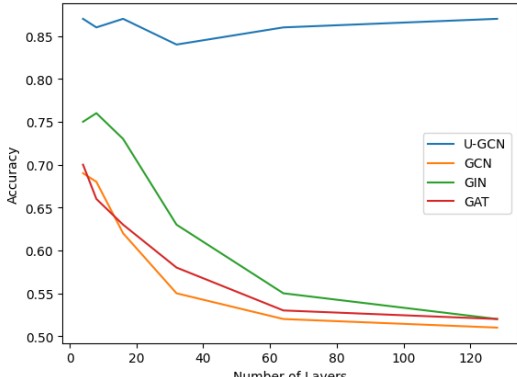

Figure 4: Test accuracies on Mutag for GCN, GIN, GAT, and a GCN with UniConv layers with increasing number of layers. Except for the unitary network, all other message passing architectures collapse to trivial accuracy levels as the number of layers increases.

positive integers. The input space $\mathbb{R}^{|D_n|}$ is indexed by group elements and convolution is performed on the regular representation of the group. Each data pair $(\boldsymbol{x}_i, y_i)_i$ is drawn as follows:

- Draw two group elements $g, g' \sim \text{Unif}(D_n)$ from the uniform distribution over the group $D_n$ without replacement.
- Set $\boldsymbol{x}_i = \boldsymbol{e}_g + \boldsymbol{e}_{g'}$ where $\boldsymbol{e}_g \in \mathbb{R}^{|D_n|}$ is a unit vector whose entries are all zero except for the entry corresponding to operation $g$ set to one.
- Set $y_i = d_{D_n}(g, g') \in \mathbb{Z}$ where $d_{D_n}(g, g')$ indicates the number of applications of the generators that need to be applied to go from $g$ to $g'$, i.e.

$$d_{D_n}(g, g') := \min_{a_1, \dots, a_{2n} \in \{s, r, r^{-1}\}} \{T : a_1 a_2 \cdots a_T g = g'\} \ . \tag{67}$$

Fig. 5 plots the performance of three different networks in this task. The vanilla network performs standard convolution in each layer where the group convolution is parameterized over elements of the generator. The residual network is the same architecture but includes skip connections after convolutions. The unitary network applies the exponential map to a skew-Hermitian version of the group convolution as outlined earlier. The unitary network is able to learn this task in fewer layers and with added stability. No hyperparameter tuning was performed in these experiments. Adam optimizer parameters were set to the default setting. All networks have a global average pooling after the last convolution layer followed by a 2 layer MLP to output a scalar. The number of channels is set to 32 for each convolution layer.

## F.4 Orthogonal Convolution

To compare orthogonal (real-valued) and unitary convolutional layers, we repeat our experiments on Peptides-func and Peptides-struct from the main text. We report the results in table 4. Edge features

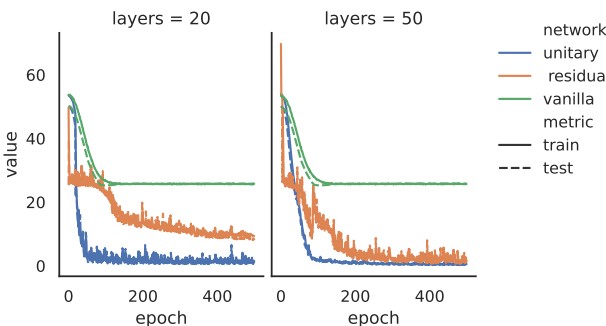

Figure 5: Training and test MAE of the distance learning task for the dihedral group. Listed as column headers are the number of convolution layers in each network. Residual and Unitary convolutional networks are both able to learn the task under default hyperparameters for the optimizer.

| METHOD | PEPTIDES-FUNC TEST AP ↑ | PEPTIDES-STRUCT TEST MAE ↓ |
|---|---|---|
| UNITARY GCN | 0.7043 ± 0.0061 | 0.2445 ± 0.0009 |
| WIDE U. GCN | 0.7094 ± 0.0020 | 0.2429 ± 0.0007 |
| NARROW O. GCN | 0.7011 ± 0.0096 | 0.2461 ± 0.0021 |
| ORTHOGONAL GCN | 0.7037 ± 0.0053 | 0.2433 ± 0.0018 |

Table 4: Comparison of GCN with Lie UniConv layers (Definition 1) with real-valued (hence orthogonal) or complex-valued (hence unitary) on Peptides-func and Peptides-struct. The Narrow Orthogonal GCN as the same width and depth as the Unitary GCN where we counted complex weights twice. Narrow Orthogonal GCN therefore has only about 285K parameters. Orthogonal GCN has the same width and depth as Wide Unitary GCN, and therefore about 490K parameters.

are not included in this ablation as no edge feature aggregator was included in the architecture. GCN with OrthoConv layers achieves performance levels similar to UniConv without using complex numbers in its weights.

## G    Experimental details

For LRGB datasets, we evaluate our GNNs using the GraphGym platform [YYL20]. In Table 1, we list reported results where available from various architectures [KW16, BL17, XHLJ18, TRRG23, HHL+23, GDBDG23, RGD+22, MLL+23, SVV+23, TRWG23]. Reported results are taken from existing papers as of May 1, 2024. Experiments were run on Pytorch [PGC+17] and specifically the Pytorch Geometric package for training GNNs [FL19].

Our implementation of unitary graph convolution does not take into account edge features. Thus, for datasets with edge features, at times, we included a single initial convolution layer using the GINE [XHLJ18] or Gated GCN [BL17] architecture which aggregates the edge features into the node features. For instances where this is done, we indicate the type of layer as an "edge aggregator" hyperparameter in the tables below.

All our experiments were trained on a single GPU (we used either Nvidia Tesla V100 or Nvidia RTX A6000 GPUs). Training time varied depending on the dataset. Peptides datasets trained in no more than 15 seconds per epoch. PascalVOC-SP took about a minute per epoch to train and COCO-SP took about 15 minutes per epoch to train. The PascalVOC-SP and two Peptides datasets are no more than 1 GB in size. The COCO dataset was much larger, about 12 GB in size. The TU and Heterophilous Node Classification datasets are all less than 1GB in size and take only a few second to train per epoch.

**Remark 12** (Parameter counting). LRGB datasets require neural networks to be within a budget of 500k parameters. For fair comparison, complex numbers are counted as two parameters each. Furthermore, to handle constraints for parameterized matrices in the Lie Algebra, we treat the

Table 5: Hyperparameters on Peptides datasets. Number of parameters are counted using the default Pytorch method which undercounts complex numbers or the methodology as stated in Remark 12. Edge aggregator indicates a single layer of the specified type which is used to incorporate edge feature data into node features.

| | Unitary GCN | | Lie Unitary GCN | |
| | PEPTIDES-FUNC | PEPTIDES-STRUCT | PEPTIDES-FUNC | PEPTIDES-STRUCT |
|---|---|---|---|---|
| lr | 0.001 | 0.001 | 0.001 | 0.001 |
| dropout | 0.1 | 0.2 | 0.1 | 0.2 |
| # Conv. Layers | 10 | 6 | 14 | 6 |
| hidden dim. | 135 | 200 | 155 | 200 |
| PE/SE | RWSE | LapPE | RWSE | LapPE |
| batch size | 200 | 200 | 200 | 200 |
| # epochs | 2000 | 500 | 4000 | 500 |
| edge aggregator | GINE | GINE | GINE | GINE |
| # Param. (see Remark 12) | 468K | 470K | 466K | 470K |
| # Param. (default Pytorch) | 282K | 305K | 464K | 305K |

Table 6: Hyperparameters on Coco and PascalVOC datasets. Number of parameters are counted using the default Pytorch method which undercounts complex numbers or the methodology as stated in Remark 12. Edge aggregator indicates a single layer of the specified type which is used to incorporate edge feature data into node features.

| | Unitary GCN | | Lie Unitary GCN | |
| | COCO-SP | PASCALVOC-SP | COCO-SP | PASCALVOC-SP |
|---|---|---|---|---|
| lr | 0.001 | 0.001 | 0.001 | 0.001 |
| dropout | 0.1 | 0.1 | 0.1 | 0.1 |
| # Conv. Layers | 12 | 15 | 12 | 15 |
| hidden dim. | 120 | 145 | 155 | 145 |
| PE/SE | None | RWSE | None | RWSE |
| batch size | 50 | 50 | 50 | 50 |
| # epochs | 750 | 1000 | 500 | 1000 |
| edge aggregator | Gated | Gated | Gated | Gated |
| # Param. (see Remark 12) | 466K | 453K | 478K | 453K |
| # Param. (default Pytorch) | 290K | 305K | 481K | 305K |

number of parameters as the dimension of the Lie Algebra (i.e. the number of parameters to fully parameterize the Lie algebra).

**Toy model: graph distance**  All networks consist of the stated number of convolution or transformer layers with feature dimension 128 followed by a global average pooling over nodes. Pooled features are then passed through a single hidden layer MLP with 128 dimension width. For networks with 5, 10 and 20 layers respectively, we set the learning rate to 0.0007, 0.0003 and 0.0001 respectively. Networks are trained with the mean average error (L1) loss. Activation functions for all networks are set to GELU apart from the unitary networks where we choose the norm-preserving activation GroupSort [TK21, ALG19] (see App. E). To ensure unitary layers are also truly norm-preserving, we do not include a trainable bias in these layers.

**Peptides**  For the Peptides experiments, the training procedure follows that of [TRRG23]. Unless otherwise stated, we use the Adam optimizer with learning rate set to 0.001 and a cosine learning rate scheduler [KB14]. Most hyperparameters were taken from [TRWG21]. For our purposes, we tuned dropout rates in the set $\{0.1, 0.15, 0.2\}$ and tested networks of layers in the set $\{6, 8, 10, 12, 14\}$ selecting the hidden width to fit within the parameter budget. For larger levels of dropout, we found that we needed to train the network for more epochs to achieve convergence. Final hyperparameters are listed in Table 6.

**COCO-SP and PascalVOC-SP**  For the COCO-SP and PascalVOC-SP experiments, our training procedure again follows that of [TRRG23]. We use the Adam optimizer with learning rate set to 0.001 and a cosine learning rate scheduler [KB14], unless otherwise stated. For our purposes, we

Table 7: Hyperparameters on the TU datasets.

| | Unitary GCN | | | | Lie Unitary GCN | | | |
| | ENZYMES | IMDB | MUTAG | PROTEINS | ENZYMES | IMDB | MUTAG | PROTEINS |
|---|---|---|---|---|---|---|---|---|
| lr | 0.001 | 0.001 | 0.001 | 0.001 | 0.001 | 0.001 | 0.001 | 0.001 |
| dropout | 0.5 | 0.5 | 0.5 | 0.5 | 0.5 | 0.5 | 0.5 | 0.5 |
| # Conv. Layers | 6 | 6 | 6 | 6 | 6 | 6 | 6 | 6 |
| hidden dim. | 128 | 128 | 128 | 128 | 256 | 256 | 256 | 256 |
| PE/SE | None | None | None | RWSE | None | None | None | RWSE |
| Edge Features | No | No | Yes | No | No | No | Yes | No |
| batch size | 50 | 50 | 50 | 50 | 50 | 50 | 50 | 50 |
| # epochs | 300 | 300 | 300 | 300 | 300 | 300 | 300 | 300 |

Table 8: Hyperparameters on the Heterophilous Graph datasets.

| | Unitary GCN | | | | | Lie Unitary GCN | | | | |
| | ROM | AMA | MINE | TOL | QUE | ROM | AMA | MINE | TOL | QUE |
|---|---|---|---|---|---|---|---|---|---|---|
| lr | 0.001 | 0.001 | 0.0001 | 0.0001 | 0.0001 | 0.001 | 0.001 | 0.0001 | 0.0001 | 0.0001 |
| dropout | 0.2 | 0.2 | 0.2 | 0.2 | 0.2 | 0.5 | 0.2 | 0.2 | 0.2 | 0.5 |
| # Conv. Layers | 8 | 8 | 8 | 8 | 8 | 6 | 4 | 8 | 8 | 4 |
| hidden dim. | 512 | 512 | 512 | 512 | 512 | 512 | 512 | 512 | 512 | 512 |
| PE/SE | LAPE | LAPE | LAPE | LAPE | LAPE | LAPE | LAPE | LAPE | LAPE | LAPE |
| batch size | 50 | 50 | 50 | 50 | 50 | 50 | 50 | 50 | 50 | 50 |
| # epochs | 2000 | 2000 | 2000 | 2000 | 2000 | 2000 | 2000 | 2000 | 2000 | 2000 |

tuned dropout rates in the set $\{0.1, 0.15, 0.2\}$ and tested networks of layers in the set $\{6, 8, 10, 12, 14\}$ selecting the hidden width to fit within the parameter budget. For larger levels of dropout, we found that we needed to train the network for more epochs to achieve convergence. Final hyperparameters are listed in Table 6.

**TU Datasets** Our experiments are designed as follows: For a given GNN model, we train on a part of the dataset and evaluate performance on a withheld test set using a train/val/test split of 50/25/25 percent. For all models considered, we record the test set accuracy of the settings with the highest validation accuracy. As there is a certain stochasticity involved, especially when training GNNs, we accumulate experimental results across 100 random trials. We report the mean test accuracy, along with the 95% confidence interval. For the TU Datasets, our training procedure and hyperparameter tuning procedure follows that described in [NHN+23]. Namely, we fix the depth and width of the network as in [NHN+23] and tune the learning rate and dropout parameters. Final hyperparameters are listed in Table 7. To ensure fairness and comparability, we conducted the same hyperparameter searches with the baseline models we compare against, but found no improvements over the numbers reported in previous papers. We therefore report their (higher) numbers instead of ours for these models.

**Heterophilous Graph Datasets** For a given GNN model, we train on a part of the dataset and evaluate performance on a withheld test set using a train/val/test split of 50/25/25 percent, in accordance with [PKD+23]. Note that we do not separate ego- and neighbor-embeddings, and hence also do not report accuracies for models from the original paper that used this pre-processing (e.g. GAT-sep and GT-sep). Our training procedure generally follows that described in the original paper [PKD+23]. We use the AdamW optimizer, stop training after no improvement in 200 steps, and including residual connections in the intermediate network layers. For the unitary GNNs, we tuned values for the dropout in $\{0.2, 0.5\}$, number of layers in $\{4, 6, 8\}$ and learning rate in $\{0.001, 0.0001\}$. To output a single number, we use a single convolution layer (SAGE convolution) to map from the higher dimensional space to a single number for each node. [PKD+23] differs in that they have an MLP in between layers; we opt for a more simple approach. Final hyperparameters are listed in Table 8. To ensure fairness and comparability, we conducted the same hyperparameter searches with SAGE and GCN, but generally found no significant differences with respect to the numbers reported in previous papers. We therefore report their usually higher numbers instead of ours for these models.

## G.1   Licenses

We list below the licenses of code and datasets that we use in our experiments.

| Model/Dataset | License | Notes |
|---|---|---|
| LRGB [DRG+22] | Custom | See here for license |
| TUDataset [MKB+20] | Open | Open sourced here |
| Heterophily Data [PKD+23] | N/A | Data is open source; no license stated in repository |
| Pytorch Geometric [FL19] | MIT | See here for license |
| GraphGym [YYL20] | MIT | See here for license |
| GraphGPS [RGD+22] | MIT | See here for license |
| Pytorch [PGC+17] | 3-clause BSD | See here for license |

