# OpenReview forum: "Unitary Convolutions for Learning on Graphs and Groups"
_NeurIPS.cc/2024/Conference — NeurIPS 2024 spotlight_

### Official Review · Reviewer_9k9v · 2024-07-05

**Soundness:** 2
**Presentation:** 2
**Contribution:** 3
**Rating:** 6
**Confidence:** 3

**Summary:**

The paper introduces two unitary graph convolution operators (UniConv and Lie UniConv), and studies their performance and ability to avoid over-smoothing even in deep Graph Neural Networks. UniConv (short for separable unitary convolution) takes the form f_{Uconv} = exp(iAt)XU (with A the adjacency matrix, i the imaginary unity, and U a unitary operator), while Lie UniConv (short for Lie Unitary Convolution) is defined as f_{Uconv} = exp(g_conv)(X) (with g_conv(X)=AXW, and W a skew-symmetric matrix). In both cases, the exponential map is approximated with a truncated Taylor series to contain complexity and avoid any form of eigen-decompositon. In experimental evaluation, GNNs implemented with UniConv achieved good performance on a variety of real-world datasets, while Lie UniConv was the only model to succeed in a toy example exhibiting long-range dependencies when compared to classic GNNs (such as GCN, GAT and Residual GCN).

**Strengths:**

The paper is generally well written and with a good introduction exhibiting the oversmoothing effect that one can observe with a classic GCN approach, and how the proposed unitary convolution can avoid that. In terms of the presentation of the methods, I didn’t have particular difficulties following the general idea of the authors, although there are some items I need to clarify with them (please see below in the weaknesses of the paper). Experiments appear overall good (though with some limitations on the Lie Uniconv method), with UniConv achieving good performance on a variety of benchmarks (including datasets showing long range dependencies and heterophilic graphs).

**Weaknesses:**

While I generally appreciated the paper (hence my slightly positive score), I’m not fully convinced by the Lie UniConv approach. The authors propose the two methods I highlighted above as a solution to avoid overfitting. As overfitting is caused by a contraction of all frequencies associated to all but the lowest frequency of the Laplace operator, converting the adjacency matrix of the provided graph into a unitary operator (as it is done in UniConv) makes sense to avoid the issue (as all eigenvalues of the obtained diffusion operator now have norm equal to 1). However, the Lie UniConv approach doesn’t go in that direction, but rather applies an exponential map on operator g_conv to obtain an orthogonal map. In this case, why the exponential of such operator is an orthogonal map is not specified in the paper, and I struggle to understand the rationale behind this approach. In this direction, in the experimental evaluation of the paper, I didn’t see any result using Lie Uniconv besides the experiment of the synthetic long rage benchmark I highlighted above. On top to this, on such benchmark, UniConv is not applied in the paper. If this was achieving comparable, or even superior, results to Lie Uniconv, then there would not be any result in the paper showing the benefit of using the Lie Uniconv method.

I would kindly ask the authors if they could clarify my doubts above, and present additional results with Lie Uniconv in the rebuttal (unless I missed them somehow in the paper).

In addition to this, I would like to point out that the results presented in Table 2 are incomplete, and while the proposed Uniconv outperforms the reported methods, it is possible to see better performing solutions on Roman E in “A critical look at the evaluation of gnns under heterophily: Are we really making progress?”. I would thus ask the authors to report all the relevant results from the paper, to ensure a fair comparison.

**Questions:**

See Weaknesses

**Limitations:**

Yes

---

> ### Author Rebuttal · Authors · 2024-08-06
>
> We thank the reviewer for their feedback and for their insightful questions. We have conducted additional experiments to answer the reviewers' main questions and will also amend the manuscript accordingly. We respond to individual points below.
>
> &nbsp;
> ___
>
> > While I generally appreciated the paper (hence my slightly positive score), I’m not fully convinced by the Lie UniConv approach. The authors propose the two methods I highlighted above as a solution to avoid overfitting. As overfitting is caused by a contraction of all frequencies associated to all but the lowest frequency of the Laplace operator, converting the adjacency matrix of the provided graph into a unitary operator (as it is done in UniConv) makes sense to avoid the issue (as all eigenvalues of the obtained diffusion operator now have norm equal to 1).
>
> This is a neat perspective on the UniConv operator that we appreciate you highlighting. Indeed, one can view this as replacing standard graph convolution which is a dissipative process with a unitary one.
>
> Let us now address the point about Lie UniConv. To continue, you write:
> > However, the Lie UniConv approach doesn’t go in that direction, but rather applies an exponential map on operator g_conv to obtain an orthogonal map. In this case, why the exponential of such operator is an orthogonal map is not specified in the paper, and I struggle to understand the rationale behind this approach.
>
> We believe there are two points here that need clarifying.
>
> First, we want to clarify what we think is a simple misunderstanding. The operator $g_{conv}$ is a linear map which lies in the Lie algebra of the orthogonal/unitary group. The matrix exponential maps from the Lie algebra to its Lie group, so the exponential map of this operator will be unitary/orthogonal meaning that $\|g_{conv}(X)\| = \|X\|$ and $g_{conv}$ is invertible. One can easily check that enforcing $W + W^\dagger=0$ enforces that the map $g_{conv}$ is in the Lie algebra.
>
> Second, there is a question about the rationale. In practice, Lie UniConv and UniConv are both unitary, but the UniConv operator acts as a tensor product $\exp(iA)\otimes W$ whereas $\exp(g_{conv})$ cannot be written in this tensor product form. The reason we separate these two is because in practice one may only want to change the message passing to be unitary leaving the feature transformation acting as before. Also, the tensor product structure makes implementation faster as one can apply feature transformation and message passing separately in sequence. Remark 2 aims to highlight these points, which we will make clearer in the revised manuscript. To return to your intuition from earlier, both of these will act as diffusion operators that are unitary. The sole difference is in how one wants to transform the feature space of the nodes. For example, one may use Lie UniConv in situations where the 'magnitude' of message passing depends on the features.
>
> We also refer the reviewer to the overall response to all reviewers for additional details about this. Since this was also a point of confusion for another reviewer, we will clarify this further in the paper (see overall response for specifics).
>
> &nbsp;
> ___
> > I didn’t see any result using Lie Uniconv besides the experiment of the synthetic long rage benchmark I highlighted above. On top to this, on such benchmark, UniConv is not applied in the paper. If this was achieving comparable, or even superior, results to Lie Uniconv, then there would not be any result in the paper showing the benefit of using the Lie Uniconv method.
>
> We thank the reviewer for sharing this feedback which was also shared by other reviewers. There are now a number of updates for this (see overall response with attached pdf). We have added a table comparing Lie UniConv to UniConv on the Peptides and TU datasets including a split for whether one parameterizes in the orthogonal or unitary group. UniConv performs slightly better though the differences are minor. Furthermore, we include a figure showing that Lie UniConv and UniConv are both equally capable of learning the toy model task.
>
> In practice, the differences between the two methods can be hard to discern. In real-world experiments, we often find that UniConv often performs slightly better since it addresses oversmoothing issues in the message passing leaving feature transformations unconstrained. Please see the overall response for more details.
>
> > I would kindly ask the authors if they could clarify my doubts above, and present additional results with Lie Uniconv in the rebuttal (unless I missed them somehow in the paper).
>
> Please see prior response. We would be happy to answer any additional questions or perform additional experiments that could help clarify the contributions of our paper.
>
> &nbsp;
> ___
> > In addition to this, I would like to point out that the results presented in Table 2 are incomplete, and while the proposed Uniconv outperforms the reported methods, it is possible to see better performing solutions on Roman E in “A critical look at the evaluation of gnns under heterophily: Are we really making progress?”.
>
> The reviewer is correct in that there are models in “A critical look at the evaluation of gnns under heterophily: Are we really making progress?” that perform better, for example on the Roman Empire dataset. However, these methodologies use different embeddings in their pre-processing. As noted in Appendix F under 'Heterophilous Graph Datasets', *"we do not separate ego- and neighbor-embeddings, and hence also do not report accuracies for models from the original paper that used this pre-processing (e.g. GAT-sep and GT-sep)"*.

---

> > ### Comment · Reviewer_9k9v · 2024-08-08
> >
> > I thank the reviewers for their response (which clarifies some of my doubts) and for the additional results listed in their attached PDF. Unfortunately, there is still no evidence on the benefit of using Lie Uniconv in the paper, which holds me back from increasing my score. Do the authors have any result comparing the speed of the two approaches perhaps?

---

> > > ### Author Response · Authors · 2024-08-08
> > >
> > > Thank you for your comment and quick feedback.
> > >
> > > Can we ask for a point of clarification? When you state `there is still no evidence on the benefit of using Lie Uniconv in the paper`, are you asking for evidence that this would outperform UniConv? As we state in the draft and in our additional experiments, we do not expect these to perform very differently. These are simply two different means of parameterizing unitary maps. They both nonetheless outperform other non-unitary message passing methods in our experiments.
> > >
> > > &nbsp;
> > >
> > > ___
> > > We are happy to provide empirical runtimes validating that unitary GCN is simply a constant factor overhead over vanilla GCN. See below.
> > >
> > > **Time in seconds for a single forward pass on a single batch of data:**
> > > ```markdown
> > > | Number of Nodes | Vanilla GCN | Spectral GCN | Gated GCN |   GPS  | Unitary GCN |
> > > |-----------------|:-----------:|:------------:|:---------:|:------:|:-----------:|
> > > | 200             |    0.0024   |    0.0054    |   0.0028  | 0.0047 |    0.0061   |
> > > | 1000            |    0.0036   |    0.0106    |   0.0064  | 0.0124 |    0.0122   |
> > > | 5000            |    0.0084   |    0.0383    |   0.0331  | 0.3009 |    0.0479   |
> > > | 25000           |    0.0733   |    0.2640    |   0.1541  | 7.1890 |    0.1894   |
> > > | 100000          |    0.2383   |    1.1645    |   1.1916  |   oom  |    1.3550   |
> > > ```
> > >
> > > The table above shows runtimes for a forward pass of a model with two convolution layers of the specified form of width 128. As expected, unitary GCN has constant factor overhead over vanilla GCN scaling similarly to other enhanced message passing schemes like spectral GCN and Gated GCN. GPS scales poorly to large graphs and runs out of memory (oom) for the largest graph of 100,000 nodes. Unitary GCN uses UniConv layers though Lie UniConv performed virtually the same in terms of runtime. All experiments are on synthetic data using the layers as implemented in pytorch geometric.
> > >
> > > Finally, we should note that we did not implement some of the efficiency improving techniques that one can implement in this line of work. For example, there exist approximations other than the Taylor approximation that may perform faster, implementations in the spectral basis can speed up unitary convolution in some instances, and there are other tricks as well (e.g. see projUNN for very fast unitary matrix updates using low rank updates). These are described in detail in the draft in appendix A and C.

---

> > > > ### Comment · Reviewer_9k9v · 2024-08-09
> > > >
> > > > Thank you again for your response. My comment was mostly dictated from the fact that I find somehow weird to have two methods proposed in a paper, when one produces consistently better performance than the other, and we have no empirical evidence of any advantage proposed by the method performing the worst. That is why I was asking for running times comparing the two methods you proposed, I was looking for proof of a good reason why one would consider implementing Lie UniConv over Uniconv.
> > > >
> > > > This said, you are right pointing out that in any case the paper proposes some methodology that outperforms prior art. I'll raise my score to 6 in view of that.

---

### Official Review · Reviewer_EVha · 2024-07-09

**Soundness:** 4
**Presentation:** 3
**Contribution:** 4
**Rating:** 8
**Confidence:** 5

**Summary:**

The paper proposes to use so-called "unitary group convolutions" in graph neural networks, with main motivation to do steps to overcome gradient collapse or explosion and oversmoothing effects in graph neural networks under equivariance constraints. This fits within a more general line of work in which unitary constraints are applied for regularizing various learning architectures.

**Strengths:**

The use of unitary constraints is a sound idea for use in GNNs, in principle.
Edit after rebuttal: It is also a novel idea, different than usual normalizations and ensuring higher nondegeneracy.

The explanations are clear and the paper is mostly well written (see "questions" part for some possible doubts).

**Weaknesses:**

Edit: after rebuttal, the main issue of originality is 100% solved, and the points in questions 3 and 5 are fully solved. The authors have proposed better experiments. This makes my concern in high part mute, the only small (and out of the scope of this work) point is that it will be great to see more verifications of oversmoothing control of beyond the (already convincing) Rayleigh Quotients preservation, which is proved in the work. As said above, this is for future work, and requires much more space, so the paper is very strong as is.
---------------------------

The originality of this approach is a bit limited, since normalizing message passing is not a very novel idea.
Doing it with unitary constraints has not been explored before for GNNs. It has been explored in other settings though.

The mathematical setup is sketchy at times, it needs better care. In particular, it is not clear if the proposed convolution is really a unitary operation (see question 3 and the crucial question 5 in "questions" section, especially).

The experiments show that normalizing is a good idea, but they don't fully convince that unitary constraints is the best way to normalize, since comparisons are done mainly against non-normalized GNNs. One has the impression that this is mainly a first exploration of a theme, and it would need more careful benchmarking, since the computational cost for implementing this particular normalization is not always negligible.

**Questions:**

What's the relative benefit compared to more naive normalizations?


Minor correctionsl/doubts:
1) lines 29-31: "it has been widely observed that group-convolutional networks suffer from instabilities as their depth increases" -- if it's widely observed, please indicate a few references where this was observed?

2) line 103: "linear convolution" -- as opposed to what other kind of convolution? isn't convolution always linear?

3) line 144: the following is an important point/issue.
"When W contains only real-valued entries, the above returns an orthogonal map" -- I don't follow this, why is that? this statement would require a proof, because it far from obvious, and in general it is false. Recall that a product of matrix exponentials is in general not the exponential of the product, so $exp (g_{\mathrm{conv}})(X) \neq e^A X e^{W^*}$ in this case.

4) I don't see what's the importance / relevance of examples 1 and 2 from Section 3. I think these are standard examples, they can be skipped or moved to the appendix.

5) lines 217-218: this probably the most important theory issue of the paper.
 "the Rayleigh quotient is invariant to unitary transformation giving a proof that unitary graph convolution avoids oversmoothing"
One can vaguely agree that Rayleigh quotient collapse may in some way be correlated to oversmoothing, but saying that Prop 6 "gives a proof that unitary convolution avoids oversmoothing" is unjustified without further proofs/details. In particular some form of a result saying that "Rayleigh quotients do not collapse implies there is no oversmoothing" is missing. Without this, the main claim of the paper hinges on "vague intuition" about RQ's and the paper is much less valuable at a theory level.

6) Definition 6 (dynamical isometry): what iis the connection of this definition with the definitions from section 3.1? This part on dynamical isometry is at present very weakly (if at all) connected to the core of the paper, and I did not understand how the authors are linking it to the main definitions. See question 3 related to this.

7) The experiments show performance on some benchmarks, but I would have been more convinced if the unitary convolution proposed method would have been compared to other simpler normalization methods which have similar effects on oversmoothing and gradient vanishing/explosion.

**Limitations:**

Main limitations are already included in the "questions" section, esp questions 3,5,7.

---

> ### Author Rebuttal · Authors · 2024-08-06
>
> We thank the reviewer for their feedback and questions. As an overall point, the reviewer made many comments about normalized convolution which we were unsure of the specific message. We kindly ask the reviewer to clarify.
> ___
> > The originality of this approach is a bit limited, since normalizing message passing is not a very novel idea. Doing it with unitary constraints has not been explored before for GNNs.
>
> Unitary layers normalize the 2-norm, but we politely disagree that they are simply a type of normalization as they also guarantee invertibility, lack of oversmoothing, stable Jacobian, etc.
>
> > What's the relative benefit compared to more naive normalizations?
>
> As stated above, we are not sure what a "naive normalization" is referring to. We believe this may refer to other interventions listed below. We stress that that unitary layers are not competing with these, but complement them: any architecture can be implemented in tandem with such techniques.
>
> 1. **batch norm**: batch norm helps normalize layer outputs, but does not have useful properties of unitary layers listed above, e.g. vanishing/exploding gradients, over-smoothing, and stability.
> 2. **regularizers and feature adjustments**: There are methods to constrain, regularize or adjust features or layer outputs to make neighboring feature vectors more dissimilar and mitigate oversmoothing. These include Dirichlet energy constraints (e.g. Zhou et al. Dirichlet energy constrained learning...), Pairnorm (Yang et al. Pairnorm...), and others. These may help over vanilla methods in addressing oversmoothing though their benefit is arguably limited -- see survey of [RBM23] and comments below.
> 3. **normalizing adjacency matrix**: One typically normalizes adjacency matrices by multiplying it by $D^{-1/2}AD^{-1/2}$ where $D$ is a diagonal matrix with $i$-th entry as the degree of node $i$. This is by default used in message passing architectures, and we use it too. This to us is a separate topic from how to perform message passing on an adjacency matrix.
> 4. **residual connections**: Though not a type of normalization, residual connections have been proposed to help avoid oversmoothing. We experimentally compare to this intervention in our experiments and also give a theoretical argument for why residual connections alone can only alleviate the problem (see Proposition 4).
>
> For more detail, consider a concurrent paper [SWJS24] using feature normalizations and residual connections. They have weaker empirical results on datasets such as Mutag and Proteins with their normalization, GraphNorm, and residual connections. We can include more details if this is what the reviewer means by "naive normalization".
>
> ___
> **Other questions/concerns:**
>
>
> > if [instabilities with depth] is widely observed, please indicate a few references where this was observed?
>
> Instabilities come in various forms and have inspired various architectectural changes, including for CNNs and GNNs (see related work). We have also included some examples. Figure 4 in the appendix shows an example of instabilities with increasing depth for several message-passing GNNs. The attached one page response also shows this issue with vanishing/exploding gradients.
>
>
> > "linear convolution" -- as opposed to what other kind? isn't convolution always linear?
>
> Linear here is linearity in the input $x$. Some convolutions are not of this form, e.g. steerable convolution acts over a set of potentially nonlinear filters.
>
> > "When W contains only real-valued entries, the above returns an orthogonal map" -- I don't follow this, why is that? this statement would require a proof, and in general it is false... $\exp(g_{conv})(X)\neq e^{A} X e^{W^*}$ in this case.
>
> We think there is a simple misunderstanding. This fact is true for the simple reason that $\exp(W)$  is real-valued whenever $W$ is real-valued (the exponential map cannot return complex numbers when its input is real-valued). So the output is always orthogonal provided $W$ meets the Lie algebra criteria of eq. (7), namely $W+W^T=0$.
> We also do not claim that $\exp(g_{conv})(X)=e^{A} X e^{W^*}$. In general, it will not take this tensor product form. Perhaps the reviewer can clarify further their concern?
>
> > I don't see what's the importance / relevance of examples 1 and 2 from Section 3. I think these are standard examples, they can be skipped or moved to the appendix.
>
> Please see general response. We will shorten section 3.2 by moving the discussion of the Fourier/spectral based methods to the appendix.
>
>
> > Saying that Prop 6 "gives a proof that unitary convolution avoids oversmoothing" is unjustified without further proofs/details...
>
> We will be more precise stating that UniConv provably avoids oversmoothing as measured by the Rayleigh Quotient or Dirichlet Energy, but we push back that this is simply "vague intuition". The Rayleigh quotient is a widespread measure of oversmoothing (see references below). Beyond this, unitary message passing is invertible and isometric, two properties that ensure signals always propogate through a network without getting "lost".
>
>
> > [vanishing/exploding gradients and dynamical isometry]: what is the connection of this definition with the definitions from section 3.1?
>
> Due to lack of space available here, please see response to reviewer vJVB asking the same question.
>
> &nbsp;
>
> **References**:
> [AL24] Roth, Andreas, and Thomas Liebig. Rank Collapse Causes Over-Smoothing and Over-Correlation in Graph Neural Networks. PMLR, 2024. \
> [RBM23] T Konstantin Rusch et al. A survey on oversmoothing in graph neural networks. arXiv:2303.10993, 2023 \
> [RCR+22] T. Konstantin Rusch et al. Graph-coupled oscillator networks. PMLR, 2022. \
> [SWJS24] Scholkemper, Michael, et al. "Residual Connections and Normalization Can Provably Prevent Oversmoothing in GNNs." arXiv preprint arXiv:2406.02997 (2024). \
> [WAWJ24] Wu, Xinyi et al. "Demystifying oversmoothing in attention-based graph neural networks." NeurIPS (2024).

---

> > ### Comment · Reviewer_EVha · 2024-08-08
> >
> > I'll slowly go through the points of your rebuttal, so this may not be my only comment (sorry).
> >
> > First of all, thanks for pointing out the difference to unitary layers, I was wrong on the novelty part, I apologize and I am grateful that you replied in detail. Reflecting upon this error of mine and your clarification, I now think that actually you could prove even stronger results about oversmoothing prevention by your approach, than the one on Rayleigh Quotients, but due to heavy review overload in 1 week time, I don't have a proposal right now, and that's anyway material for future work.
> >
> > About my correction/doubt number 3: when $W$ is an antisymmetric real matrix, $\exp(W)$ is orthogonal, we agree this far.
> >
> > But in line 144 I'm concerned that you say "the above outputs an orthogonal map". In that comment, you refer to $exp(W)$ or to $f_{Uconv}(X)$? Given that $f_{Uconv}$ has a "U" in it, and given this paper's title, I assume it's $f_{Uconv}$.
> >
> > How does "exponential of real-valued antisymmetric implies orthogonal" apply to $f_{Uconv}(X)=\exp(g_{conv}(X))=\exp(AXW)$ ? are you using the fact that $AXW$ is antisymmetric real-valued too if $W$ is? I don't follow.. I'd like to see a proof of the map being orthogonal.
> >
> > Thanks for the detailed rebuttal so far.

---

> > > ### Author Response · Authors · 2024-08-08
> > >
> > > Thank you for your quick response and especially for recognizing the novelty of our work. We would be happy to hear about your thoughts on how we can prove stronger results on oversmoothing if you are willing to share it.
> > >
> > > ___
> > > Regarding your question, yes we are referring to $f_{Uconv}$. Let us go through this in more detail.
> > >
> > > As a reminder, we have that $f_{Uconv}(X)=\exp(g_{conv})(X)$ and $g_{conv}(X) = AXW$ and for an undirected graph $A = A^\dagger$ (for directed graphs, see Appendix D for proper handling of this). Furthermore, we constrain $W$ so that $W + W^\dagger=0$. Treating inputs and outputs as vectors, we can equivalently view $g_{conv}$ as acting on a vector rather than a matrix:
> > >
> > > $$ g_{conv}(X) = AXW \leftrightarrow vec( g_{conv}(X) ) = (A \otimes W^T) vec(X) , $$
> > > where we vectorize $X \in \mathbb{C}^{a \times b}$ with the command $vec(X) \in \mathbb{C}^{ab}$ that stacks columns on top of each other. Now, note that
> > >
> > > $$ (A \otimes W^T)  + (A \otimes W^T) ^\dagger = A \otimes (W + W^\dagger)^T = 0, $$
> > >
> > > and thus $(A \otimes W^T)$ is in the Lie Algebra of the Unitary/Orthogonal group. Finally, we now have
> > > $$vec(\exp(g_{conv}(X))) = \exp(A \otimes W^T) vec(X),$$
> > > and thus $\exp(A \otimes W^T)$ is in the Unitary/Orthogonal Lie group.
> > >
> > > Hopefully, viewing it this way in the "vector sense" clarifies things. Let us know if additional details are requested. We will clarify this in the updated manuscript.

---

> ### Comment · Reviewer_EVha · 2024-08-08
>
> Ok, thanks for the detailed reply. I hope it won't bother you if I continue a bit with the clarifications on the same topic (sorry if it does):
>
> 1) (minor point) So if $A=A^T$ and $W=-W^\dagger$ then these are square matrices, and then $X$ must be a square matrix, right? Or how can I compute $\exp(AXW)$ if $AXW$ is not a square matrix? (I'm confused by the notation $X\in \mathbb C^{a\times b}$, is it that necessarily $a=b$?)
>
> 2) Assume now that $A=Id$, and $X,W$ are real-valued square matrices of equal dimension $n$, of which $W=-W^T$, to simplify things. Say $A=Id$, and $W=\left[\begin{array}{cc}0&-1\\\\ 1&0\end{array}\right]$ and $X=\left[\begin{array}{cc}0&1\\\\ 0&0\end{array}\right]$. Then $AXW=\left[\begin{array}{cc}1&0\\\\ 0&0\end{array}\right]\neq -(AXW)^T$ right? Then $\exp(AWX)$ is not unitary or orthogonal. I am confused, so maybe clarifying what I misunderstood with this example can help.

---

> ### Author Response · Authors · 2024-08-08
>
> No problem; we are happy to clarify. Let us answer your questions in order.
>
>
> 1. Formally, unitary transformations must be invertible so the input and output dimension must be equal. Thus the row and column dimensions of $W \in \mathbb{C}^{b\times b}$ must be equal and similarly for $A \in \mathbb{C}^{a \times a}$. However, this does not mean that $X$ has equal row and column dimension. The map $f_{Uconv}$ maps matrices of size $a \times b$ to matrices also of size $a \times b$ so dimensionality is preserved in this sense.
> As noted in the draft though, this does not limit implementation: for $W$ with arbitrary row and column dimension, one can also work within the Stiefel manifold or even more simply pad inputs and outputs so that dimensionalities are equal to actually implement things in practice (see appendix D).
> 2. It does not need to hold that $AXW =- (AXW)^\dagger$.  In fact, this will not hold true in general since the row and column dimensions of $X$ are not equal so this equation formally may not even make sense. What does hold true however is that the linear operator $g_{conv}$ is in the Lie algebra. As a reminder, exponential map is equal to
> $$\exp(g_{conv})(X) = \sum_{k=0}^\infty \frac{g_{conv}^{(k)}(X)}{k!} = \sum_{k=0}^\infty \frac{A^kXW^k}{k!}. $$
> Viewed in the vectorized form from our previous comment this is equivalent to:
> $$vec[\exp(g_{conv})(X)] = \sum_{k=0}^\infty \frac{(A \otimes W^T)^k}{k!} vec[X],$$
> so as stated earlier, one can check that $A \otimes W^T$ is in the Lie algebra to ensure unitarity.
>
>
> Please let us know if we can clarify anything further.

---

> > ### Comment · Reviewer_EVha · 2024-08-08
> > **All doubts clarified, thanks**
> >
> > I understand now. I was interpreting $g_{conv}^k(X)$ as $(g_{conv}(X))^k$ and now that I noticed the difference everything is clear.
> >
> > My main concerns have been all lifted this is the case for concern for novelty (which is now "reverted": I believe the paper has strong novelty), and the topics of questions 3 and 5 (technical points).
> >
> > The concern that Rayleigh Quotients may be only one of several possible measures for oversmoothing is still valid for me, but I think it can/should be considered in future work, and due to understanding the difference between normalizing 2-norm and imposing unitary transformations, I am confident in the result being "fortifiable". What I mean is mainly to prove nondegeneracy according to more of the used metrics for oversmoothing.
> >
> > Given the above, I think it's fair to raise my score to 8.

---

> > > ### Author Response · Authors · 2024-08-09
> > >
> > > Thank you once again for your valuable feedback and the constructive discussion. We appreciate your decision to raise your score to an 8.
> > >
> > > The score change does not appear on our side. Would you mind verifying that the updated score has been recorded in the system?

---

### Official Review · Reviewer_feJe · 2024-07-12

**Soundness:** 4
**Presentation:** 3
**Contribution:** 4
**Rating:** 8
**Confidence:** 4

**Summary:**

In this paper the authors propose a mathematically consistent treatment of unitary convolutions in the context of neural networks defined on graphs and groups. In the graph neural network context, the main idea is to transform a standard linear convolutional layer $f_\text{conv} = \mathbf{AXW}$ with $\mathbf{A}$ the adjacency matrix, $\mathbf{X}$ the signal input matrix and $\mathbf{W}$ the parameterize into a linear layer $f_\text{Uconv}$ that is unitary (norm preserving), taking into account both the matrices $\mathbf{A}$ and $\mathbf{W}$ (previous literature focus on $\mathbf{W}$ or perform this in a less natural way (with worse computational complexity) [QBY24]. The main idea revolves around the following two steps: (1) project the operation into matrix Lie algebra (e.g., the space of skew-symmetric matrices) (2) apply the exponential map on the projected operation to obtain an unitary operation. In the graph setting the authors do this in two ways: the first is called "Separable unitary graph convolution (UniConv)" and takes the form $f_{\text{Uconv}} (\textbf{X}) = \exp(i\textbf{A}𝑡)\textbf{X}\textbf{U}, \textbf{U} \textbf{U}^{\dagger} = \textbf{I}$. The second is called "Lie orthogonal/unitary graph convolution (Lie UniConv)" and is defined by applying an exponential map on a convolution with skew-symmetric weight matrices in Lie algebra, having the advantage of being parameterizable over the reals only. The authors then discuss how to approximate the exponential map (they resort to simple Taylor truncation).  Subsequently, the authors develop the theory in the case of general groups and show to ways of obtaining unitary convolutions: the first case generalizes the two methods defined for graphs, by first skew-symmetrizing the equivariant map and then applying exp; the second is defined as a (block-diaognal) convolution in the Fourier domain followed by inverse transform. Following the authors show basic properties of the proposed method (invertibility, isometry and equivariance) and prove favorable results which mainly follow by isometry (avoiding oversmoothing, avoiding vanishing/exploding gradients). In the experimental section, the authors perform experiments on toy ring graph (on Lie UniConv) and the standard benchmarks LRGB and Heterophilous Graph Dataset (on UniConv) (also showing group convolution in Appendix on Dihedral group $D_n$).

**Strengths:**

- Well founded mathematical formalism, defining unitarity in message passing graph neural networks in a natural way.
- Strong performance over other message passing algorithms, comparable performance with respect to graph attention based models. On very large graphs attention could be prohibitive so UConv could perform well in that situation.
- The paper has an extensive and complete literature overview, comparing with related literature in very detailed manner (the Appendix on related work relative to unitary convolutions is a very good addition and clearly let us understand the contribution).
- Theoretical analysis on oversmoothing and vanishing/exploding gradients shows the advantage on adopting UConvs in a quantitative way.

**Weaknesses:**

- There aren't many experiments on Lie UniConv. From what I understand, the only experiment with this layer is the graph ring experiment in Figure 2. Also it would have been interesting to see comparison between the two type of convolutions (UniConv vs Lie UniConv), with the more realistic datasets, in order to asses which one performs better in practice.

- The part on unitary maps in Fourier basis feels somehow detached from the general discourse of the paper. I believed the author added it for completeness but (1) the theory to understand it is relatively advanced (had to study externally some representation theory to understand well the definition of Fourier transform on finite groups); a short primer on representation theory in the Appendix, similar to the one for Lie groups / algebras, would have been of great help. (2) There are no experiments in this setting. (3) some parts are a little bit shaky, for example in: ```One can also generally implement convolutions in the (block diagonal) Fourier basis of the graph
or group (Algorithm 2). Here, one employs a Fourier operator which block diagonalizes the input into its irreducible representations or some spectral representation.```, if one considers using the eigendecomposition of some graph Laplacian as a spectral basis, the (graph) Fourier transform is not given by a set of matrices varying over irreps but a signal of the same dimensionality of the input function. I do not think one can use the same approach there.

- Sometimes authors leave facts implicit and the writing could be improved in key places. For example I had difficulty understanding why Lie UniConv returns an unitary map before looking at the tensor product version of $g_{\text{conv}}$ (Eq. (36), Proposition 6). Also the statements in Propositions 4 and 7 are not really clear from a first reading, some more intuition about the statements should be given.

**Questions:**

- I don't see how the graph convolutions defined are particular cases of the group convolutions described. Is it the symmetric group S(n) acting on the graph?
- Line 122: Why UniConv has a "tensor product nature"? From what I understand the tensor product is more related to the Lie UniConv because you can write it as $\exp(\mathbf{A} \otimes \mathbf{W}^T)$.
- Line 186: "We set input/output representations to be equal so that the exponential map of an equivariant operator is
itself equivariant": Assuming this does not limit the class of groups or transforms we can use? Not all groups have irreps of same dimensions. Or we say that the transforms are equivariant only in this sense? Also why exponential map is equivariant only if the input output representations are equal?
- Line 212: is (I − A) a type of graph Laplacian?
- Line 272: Why authors don't compare to new architectures as well on this task, like in Table 1? Also I see in Appendix that GPS it is performing pretty well w.r.t. Unitary GCN but it was not put in this graph (maybe to show better improvements?).
- Line 763: I think authors should say that in case of Algo 2 this follows because Fourier transform is unitary. It is not clear if and why this should be the case in finite group setting.
- Line 766: "For unitary convolution in the Fourier domain (Algorithm 2), equivariance follows from convolution theorems where linear operators appropriately applied in the block diagonal Fourier basis are equivariant". This is too succinct and should be more expanded formally.
- Line 821: missing trace on following lines and last line should not be put there ($= R_{\mathcal{G}(\mathbf{X})}$)

**Limitations:**

The authors have discussed limitations of their contribution adeguatly.

---

> ### Author Rebuttal · Authors · 2024-08-06
>
> We thank the reviewer for their positive comments and going through the paper in detail. There are many insightful questions and points of feedback that we will incorporate into the draft.
> ___
>
> > There aren't many experiments on Lie UniConv... it would have been interesting to see comparison between the two
>
> We agree with the reviewer on this point and have included many comparisons in the global response and attached one page pdf there. This includes experiments for Peptides and TU datasets. In general, the empirical differences are rather minimal.
>
> > The part on unitary maps in Fourier basis feels somehow detached from the general discourse of the paper... the theory to understand it is relatively advanced... a short primer on representation theory in the Appendix... would have been of great help. There are no experiments in this setting.
>
> We aimed to show that Fourier space can often speed up operations and simplify convolution as observed in many works. Since this is tangential to the main points, we'll move it to the appendix with a detailed background. In the main text, we'll briefly note that a Fourier basis methodology exists.
>
> > The part on unitary maps in Fourier basis feels somehow detached... if one considers using the eigendecomposition of some graph Laplacian as a spectral basis, the (graph) Fourier transform is not given by a set of matrices varying over irreps but a signal of the same dimensionality of the input function.
>
> We believe there is a small misunderstanding here. One can diagonalize the adjacency matrix of an undirected graph and perform convolution over the spectral basis as in spectral graph convolution. When the input and output is a single channel, this can be treated as block diagonal operations, but the blocks are trivially scalars  ($1 \times 1$ matrices). More generally, the block size will depend on the input/output channel dimensions. We will provide a lemma explicitly showing this in the more general case for arbitrary graphs/groups.
>
> > Sometimes authors leave facts implicit... I had difficulty understanding why Lie UniConv returns an unitary map before looking at the tensor product version of (Eq. (36), Proposition 6). Also the statements in Propositions 4 and 7 are not really clear from a first reading, some more intuition about the statements should be given.
>
> We agree this was slightly confusing on a first read. We will revise these passages for clarity in the updated manuscript.
>
> &nbsp;
> ___
> **Questions:**
>
> > I don't see how the graph convolutions defined are particular cases of the group convolutions described. Is it the symmetric group S(n) acting on the graph?
>
> To map graph convolution to this case where $conv_G(X) = \sum_i T_i X W_i$, simply set $T_i= A^i$ or the $i$-th power of the (potentially normalized) adjacency matrix. We will explicitly state this after the equation in the main text.
>
> > Line 122: Why UniConv has a "tensor product nature"? From what I understand the tensor product is more related to the Lie UniConv because you can write it as $\exp(A \otimes W^T)$.
>
> The reviewer is right that Lie UniConv is parameterized in a tensor product fashion, but crucially, the output is not a tensor product: there may not exist matrices $M_1, M_2$ such that $\exp(A \otimes W^T) \neq M_1 \otimes M_2$. In contrast, UniConv is always of the form $\exp(iA)\otimes W^T$ and one can apply the feature and message passing transformations separately and in sequence. We tried to capture this in remark 2, but will clarify in the main text.
>
> > Line 186: "We set input/output representations to be equal so that the exponential map of an equivariant operator is itself equivariant": Assuming this does not limit the class of groups or transforms we can use? Not all groups have irreps of same dimensions...
>
> This statement is written confusingly and we will clarify. What we meant to say is that the input/output representations need to act on the same vector space. For a unitary transformation to be well defined, it needs to be invertible and isometric so the input and output vector space are must have the same dimension. One can generalize beyond these assumptions, but it seems unnecessary for our purposes.
>
> > Line 212: is (I − A) a type of graph Laplacian?
>
> That is correct.
>
> > Line 272: Why authors don't compare to new architectures as well on this task, like in Table 1? Also I see in Appendix that GPS it is performing pretty well w.r.t. Unitary GCN but it was not put in this graph (maybe to show better improvements?).
>
> We show GPS for sake of completeness, but did not include it in the main plots because it is not a message passing architecture. The goal of the toy model task was to see which architectures can learn long range dependencies with local operations. For transformers, the notion of long range dependency is not as well defined.
>
> Is there another method that the reviewer feels we should compare to? We are happy to include it.
>
> > Line 763: I think authors should say that in case of Algo 2 this follows because Fourier transform is unitary.
> > Line 766: "For unitary convolution in the Fourier domain (Algorithm 2), equivariance follows from convolution theorems...". This is too succinct and should be more expanded formally.
>
> Yes, we will say that and expand on this. Thank you for pointing it out.
>
> > Line 821: missing trace on following lines and last line should not be put there ($=R_G(X)$)
>
> We think there is a misunderstanding. The following lines have the notation $vec(X)^\dagger ... vec(X)$ which is an equivalent way of writing the trace as an inner product between vectors since $Tr(A^\dagger B)= vec(A)^\dagger vec(B)$ where $vec$ vectorizes the matrices. We will use inner product angle bracket notation to make this clearer. The last line also needs the normalization by the Frobenius norm which we will include as a separate line. Thank you for pointing this out.

---

> > ### Comment · Reviewer_feJe · 2024-08-12
> >
> > I thank the authors for addressing different points such as additional experiments on Lie UniConv and improving the clarity of the paper where I have pointed out different misunderstandings. The papers is a solid contribution and should definitely be accepted.

---

### Official Review · Reviewer_vJVB · 2024-07-12

**Soundness:** 2
**Presentation:** 1
**Contribution:** 2
**Rating:** 4
**Confidence:** 3

**Summary:**

This paper introduces convolutional layers for GNNs that are based on group-convolutions. Some of the proposed layers have interesting theoretical properties such as avoiding oversmoothing or vanishing gradients. Experimentally, these layers are able to aggregate information across long distances and achieve strong results on real-world benchmarks.

**Strengths:**

- **(S1 - Novelty)** The contributions in this paper seem novel.

- **(S2 - Significance)** From a theory perspective, some of the proposed operations have interesting theoretical properties such as that they can avoid oversmoothing or vanishing gradients.

- **(S3 - Significance)** The authors perform extensive experiments on real-world datasets and achieve strong results on most of them.

**Weaknesses:**

- **(W1 - Clarity and Significance)** The reader is not guided through the paper and it is difficult to understand the flow of ideas. I will give some examples of this:

   - In Section 3.1 (Lie) UniConv layers are introduced which is the main contribution of the pape. However, to me it was not clear that this is the central definition.

   - In the following Section 3.2, the paper generalizes the setting of unitary convolutions and gives several examples. Yet, to me it is unclear what this section intends to communicate and how it is relevant to the rest of the paper.

   - Another example of this is the “Vanishing/Exploding gradients”-part of Section 4, here dynamical isometry is defined as a measure as a property to analyze exploding gradients and an example is given that a combination UniConv together with a yet completely undefined activation function (GroupSort) is “perfectly dynamically isometric”. This raises many questions to the reader: Why is an example important enough to warrant the definition of Dynamical isometry? How does this relate to the main contribution of the paper? Are vanishing / exploding gradients a problem with GNNs? What is GroupSort and is it actually used in practice?

- **(W2 - Clarity)** The experiment section is difficult to read and does not properly explain the experiments or the design decisions. For example, for the graph distance dataset the model is not mentioned in the text. In this case, the model is GCN with the novel Lie UniConv layer (Figure 2). However, it is not explained why the Lie UniConv layers are used and not the UniConv Layers. Similarly, for the experiments on LRGB UniConv layers are used but it is not explained why Lie UniConv layers are not tried.

**To sum up.** To me it seems like this could be a good paper with interesting ideas and solid experiments. However, the writing significantly holds it back and I think more time is required to allow  the authors to rewrite their paper. Furthermore, I think that there is a lot of useful information hidden in the appendix and it might make sense to focus the paper on the main idea (Section 3.1) and move other parts to the appendix to make more space.  As I do believe that this paper requires significant changes to the text I thus vote to reject.

**Questions:**

Most of my questions are already part of the weakness section.

- **(Q1)** (Lie) UniConv are implemented as a $k$-th taylor approximation. What is the value for $k$? It seems to me like the ability to model long-range interactions is primarily dependent the value of $k$ and the number of layers $L$, so knowing this value seems important.

**Limitations:**

Yes.

---

> ### Author Rebuttal · Authors · 2024-08-06
>
> We thank the reviewer for praising the novelty and theoretical results in our work. We have read their concerns about the format and writing of the paper, and hope to address them below.
> ___
> > In Section 3.1 (Lie) UniConv layers are introduced... it was not clear that this is the central definition.
>
> Can the reviewer expand on what they mean by central definition? From our understanding, the definitions of UniConv operations are central to the paper.
>
>
> > The paper generalizes the setting of unitary convolutions... it is unclear what this section intends to communicate...
>
> Our goal was two-fold: to show that unitary convolution on graphs is a simple method applicable to general symmetric domains, and to connect it to previous CNN implementations, demonstrating how to generalize beyond graph convolution, such as applying unitary convolution to hypergraphs or other message-passing types. That being said, we understand the reviewer's concerns. We will shorten this section in the paper focusing specifically on the points above. Please see our general response for details.
>
> > Dynamical isometry is defined as a measure as a property to analyze exploding gradients and an example is given that a combination UniConv together with a yet completely undefined activation function (GroupSort) is “perfectly dynamically isometric”... Why is an example important enough to warrant the definition of Dynamical isometry?
> > Are vanishing / exploding gradients a problem with GNNs?
>
> Vanishing/exploding gradients are a general problem with any deep or recurrent network and occur because norms of the states at intermediate layers grow or decay exponentially with depth (see e.g. [LMTG19], [LMGK21]). GNNs are no different from deep networks in that they also have weights initialized randomly causing norms to grow or decay as a product of the number of layers. We found that standard GNNs cannot train at very large depths (see figure 4 showing standard message passing schemes do not train well at around 40 layers for example). We also show an example of a standard GCN featuring vanishing/exploding gradients in the one page response pdf (UniConv does not have this issue in contrast).
>
> For scalability to many layers, stable information and gradient propagation is essential. Dynamical isometry offers a rigorous framework to study this stability, supported by experiments and theory across various architectures. We aim to show that GNNs also exhibit the stability properties described by dynamical isometry.
>
>
>
>
> > What is GroupSort and is it actually used in practice?
>
> Groupsort is a nonlinearity that splits the input vector into pairs of numbers and sorts the pairs (see eq. 58). It is a nonlinearity that preserves isometry and used in the adversarial learning literature to get provable adversarial bounds via unitary architectures. E.g., see its use in Trockman & Kolter Orthogonalizing convolutional layers with the cayley transform.
>
>
>
> > For the graph distance dataset the model is not mentioned in the text. In this case, the model is GCN with the novel Lie UniConv layer (Figure 2). However, it is not explained why the Lie UniConv layers are used and not the UniConv Layers. Similarly, for the experiments on LRGB UniConv layers are used but it is not explained why Lie UniConv layers are not tried.
>
> Please see global response and additional figures/tables for a full answer to this. In summary, this choice did not make a significant difference. For experiments where networks were not very deep, we found that enforcing unitarity in the message passing was helpful and unitarity was not needed in the feature transformation.  For example, UniConv performs slightly better in LRGB though little difference is observed.
>
>
>
>
>
> > To me it seems like this could be a good paper with interesting ideas and solid experiments...I think more time is required to allow the authors to rewrite their paper... there is a lot of useful information hidden in the appendix and it might make sense to focus the paper on the main idea (Section 3.1) and move other parts to the appendix to make more space.
>
> If we understand the reviewer correctly, they have requested that we focus more on section 3.1, defer parts of section 3.2 to appendix, and make more space for clarifying some points such as when to use Lie UniConv vs. UniConv. We will make these changes in the revised manuscript. Are there other parts of the paper that the reviewer feels should be moved from the appendix to the main text or vice versa?
>
> We are hesitant to make significant changes to the writing and format given this concern was not shared by all the reviewers, but are of course open to more feedback.
>
> > (Lie) UniConv are implemented as a k-th taylor approximation. What is the value for k?... knowing this value seems important.
>
> We found in all our experiments that $k=10$ suffices. The error in the Taylor approximation exponentially decreases with $k$. This value is in line with that in other papers (e.g. see [SF21]). We will state this in the main text.
>
> ___
> &nbsp;
>
> **Dynamical Isometry References:**
>
> [SMG13] Saxe et al. Exact solutions to the nonlinear dynamics of learning in deep linear neural networks. arXiv:1312.6120, 2013 \
> [XBSD+18]  Xiao et al. Dynamical isometry and a mean field theory of cnns: How to train 10,000-layer vanilla convolutional neural networks. ICML, 2018 \
> [PSG18] Pennington et al. The emergence of spectral universality in deep networks. PMLR, 2018
>
> &nbsp;
>
> **Vanishing/ Exploding Gradient Problems with GNNs:**
> In addition to above, see also:
>
> [LMTG19] Li et al. Deepgcns: Can gcns go as deep as cnns?. Proceedings of the IEEE/CVF international conference on computer vision, 2019. \
> [LMGK21] Li et al. Training graph neural networks with 1000 layers. PMLR, 2021.

---

> > ### Comment · Reviewer_vJVB · 2024-08-13
> >
> > Dear authors, I am sorry for the delay, I submitted a reply with the wrong reader-group. You should now be able to see my reply above.

---

> ### Comment · Reviewer_vJVB · 2024-08-12
>
> I thank the authors for their thorough rebuttal. The additional experiments do strengthen the paper and including the results for both architectures does in my opinion simplify the experiment section.
>
> > Can the reviewer expand on what they mean by central definition? From our understanding, the definitions of UniConv operations are central to the paper.
>
> Yes, they are. However, for some reason this was not immediately clear for me when reading this section. Maybe a bit more guiding of the reader is required.
>
>
> > If we understand the reviewer correctly, they have requested that we focus more on section 3.1, defer parts of section 3.2 to appendix, and make more space for clarifying some points such as when to use Lie UniConv vs. UniConv. We will make these changes in the revised manuscript. Are there other parts of the paper that the reviewer feels should be moved from the appendix to the main text or vice versa?
>
> You understand me correctly, I think that moving 3.2 to the appendix is a good idea. This  should allow you to focus more on your essential results (key definitions + theorems/propositions) and intuitively explain what they mean and why they matter.
>
> > We are hesitant to make significant changes to the writing and format given this concern was not shared by all the reviewers, but are of course open to more feedback.
>
> I agree with this sentiment. While I do think that my issues with the presentation / clarity persist (and this is somewhat mirrored by feJe), I cannot ignore that this does not seem to be an issue for the other reviewers. Furthermore, upon a further look I have to concede that this paper does contain many non-trivial theoretical insights. I will slightly increase my score but adjust my confidence downward.

---

### Official Review · Reviewer_9r1r · 2024-07-12

**Soundness:** 3
**Presentation:** 3
**Contribution:** 3
**Rating:** 7
**Confidence:** 3

**Summary:**

This paper proposed a new convolution operator UniConv on graphs and general tasks. Both graph convolution and general form convolution form are provided based on the group theory. With the toy experiment, it shows the advantages of the proposed method clearly. Based on both theoretical and experimental analysis, the proposed network is effective in long range graphs and heterophilic graphs.

**Strengths:**

1. Theoretical analysis is conducted to show the effectiveness in handling oversmoothing and oversquashing issue.

2. The toy analysis is easy to understand and clearly shows the advantage.

**Weaknesses:**

1. Though the general form of convolution is provided. It is not evaluated in experiment. I will suggest putting these content in the appendix in this case. Otherwise, it is better to provide the corresponding experimental results.

2. The computational cost is higher compared to other methods, but there are no experimental results to show the important details.

**Questions:**

See weakness.

---

> ### Author Rebuttal · Authors · 2024-08-06
>
> We thank the reviewer for their positive comments and feedback. We respond to their questions and comments below.
>
> &nbsp;
> ___
> > Though the general form of convolution is provided. It is not evaluated in experiment. I will suggest putting these content in the appendix in this case. Otherwise, it is better to provide the corresponding experimental results.
>
> We thank the reviewer for pointing this out. Our primary goal was to show a general recipe for constructing equivariant unitary/orthogonal maps, which allows for introducing more general convolution on finite groups beyond just graph convolution. Our main application are GNNs, hence, much of the experiments are focused on the graph setting. We provide experimental results for another instance of the general case, namely group convolution over the dihedral group, in Appendix E.3. We note that previous literature has studied unitary/orthogonal CNNs which also fit within this framework. We discuss and reference these works in the extended related works (see Appendix A.1). We will add more explicit pointers to these sections in the main text.
>
> &nbsp;
> ___
> > The computational cost is higher compared to other methods, but there are no experimental results to show the important details.
>
> The unitary and orthogonal layers have a constant factor overhead over standard message passing. As stated in the paper, the runtime is $O(KnD)$ where $K$ is the approximation in the exponential map, $n$ is the number of nodes, and $D$ is the maximum degree of the graph. Since the approximation error is exponentially small in $K$ (see Appendix C), we found that setting $K$ to be a constant (around 10 for example) sufficed in all experiments. We will specify this in the main text in the revised manuscript.
>
> Note, that this is more efficient than many graph algorithms such as graph transformers which are quadratic scaling in $n$ and other proposed (near) unitary graph message passing algorithms which also scale poorly in $n$ and $E$.
> The computational cost is detailed further in Appendices A and C. We should also note that this factor $K$ overhead is in some sense unavoidable if one wants invertibility and isometry as detailed in Proposition 4.

---

### Author Rebuttal · Authors · 2024-08-06

We thank all the reviewers for their insightful comments, recommendations, and feedback.

&nbsp;

There were some common themes in the reviews which we want to address in an overall response here. We have also included a one page pdf containing plots and tables which help address reviewers' concerns.

 1. **Lie UniConv experiments:**
Reviewers requested additional experiments comparing Lie UniConv to UniConv. We originally did not include these as they performed similarly, and apologize for the oversight. We include in the one page pdf experiments comparing Lie UniConv and UniConv (both with real-valued orthogonal and complex-valued orthogonal/unitary matrices) for the Peptides experiments and TU datasets. Overall, performance is very similar for Lie UniConv and UniConv.
We also include a figure showing that UniConv and Lie UniConv are equally effective at learning the toy model task of graph distance.
All results will be included in the revised version of the paper.

2. **Vanishing/exploding gradients:**
Reviewers vJVH and EVha questioned the relevance and importance of vanishing/exploding gradients. In response, we have included a simple plot showing vanilla GCNs are no different than other deep networks and can suffer from this issue. This plot also shows unitary GCNs avoid this issue. This plot was created using standard initialization and settings of the graph convolution layer in pytorch Geometric. Of course, vanishing/exploding gradients are a common problem accross deep networks of all kinds, and unitary layers are not the only way to help alleviate this problem.

3. **Generalized and Fourier basis convolution:**
Section 3.2 generalizes unitary graph convolution to convolution over arbitrary finite groups and in the Fourier/spectral basis. While we believe that this more general perspective is instructive to the reader in that it gives background on other attempts at unitary/orthogonal convolution, we agree that the discussion of the Fourier/spectral based methods in the main paper distracts from the main contributions of the paper.
Based on your detailed comments and feedback, we have decided to make the following changes to the presentation: We will defer the description of Fourier/spectral basis convolution to the appendix and include a richer background into representation theory to make it more accessible to the reader. We will shorten the discussion in the main text to a paragraph to informally present potential advantages of implementation in the Fourier domain and note that we did not implement this in our work. These changes will give us space to include additional experimental results for Lie UniConv.

&nbsp;

***Please see attached one page pdf with additional tables and plots below.***

---

### Decision · Program_Chairs · 2024-09-25

**Decision:**

Accept (spotlight)

**Comment:**

This paper introduces unitary convolutional layers for Graph Neural Networks (GNNs), with the primary aim of mitigating issues like gradient collapse, explosion, and over-smoothing in GNNs. The experimental results demonstrate that these layers can effectively aggregate information across long distances and achieve strong performance on real-world benchmarks.

The paper was well received by the reviewers, who appreciated the formalism and theoretical analysis of the proposed operators. The initial scores generally improved after the authors' rebuttal, which clarified some misunderstandings and provided additional experimental results.

The main limitation pointed out by the reviewers concerns the writing and organization of the paper. This is the main reason for the “borderline reject” score from one of the reviewers, who, however, decided to lower his confidence score. The authors have proposed addressing these issues with revisions that should be applied in the camera-ready version. Additionally, the introduction of two unitary convolution operators caused some confusion, and the authors should better motivate their inclusion.